# SPACE AND TIME CONTINUOUS PHYSICS SIMULATION FROM PARTIAL OBSERVATIONS

**Steeven Janny**
LIRIS, INSA Lyon, France
`steeven.janny@insa-lyon.fr`

**Madiha Nadri**
LAGEPP, Univ. Lyon 1, France
`madiha.nadri-wolf@univ-lyon1.fr`

**Julie Digne**
LIRIS, CNRS, France
`julie.digne@cnrs.fr`

**Christian Wolf**
Naver Labs Europe, France
`christian.wolf@naverlabs.com`

## ABSTRACT

Modern techniques for physical simulations rely on numerical schemes and mesh-refinement methods to address trade-offs between precision and complexity, but these handcrafted solutions are tedious and require high computational power. Data-driven methods based on large-scale machine learning promise high adaptivity by integrating long-range dependencies more directly and efficiently. In this work, we focus on fluid dynamics and address the shortcomings of a large part of the literature, which are based on fixed support for computations and predictions in the form of regular or irregular grids. We propose a novel setup to perform predictions in a continuous spatial and temporal domain while being trained on sparse observations. We formulate the task as a double observation problem and propose a solution with two interlinked dynamical systems defined on, respectively, the sparse positions and the continuous domain, which allows to forecast and interpolate a solution from the initial condition. Our practical implementation involves recurrent GNNs and a spatio-temporal attention observer capable of interpolating the solution at arbitrary locations. Our model not only generalizes to new initial conditions (as standard auto-regressive models do) but also performs evaluation at arbitrary space and time locations. We evaluate on three standard datasets in fluid dynamics and compare to strong baselines, which are outperformed both in classical settings and in the extended new task requiring continuous predictions.

## 1 INTRODUCTION

The Lavoisier conservation principle states that changes in physical quantities in closed regions must be attributed to either input, output, or source terms. By applying this rule at an infinitesimal scale, we retrieve partial differential equations (PDEs) governing the evolution of a large majority of physics scenarios. Consequently, the development of efficient solvers is crucial in various domains involving physical phenomena. While conventional methods (e.g. finite difference or finite volume methods) showed early success in many situations, numerical schemes suffer from high computational complexity, in particular for growing requirements on fidelity and precision. Therefore, there is a need for faster and more versatile simulation tools that are reliable and efficient, and data-driven methods offer a promising opportunity.

Large-scale machine learning offers a natural solution to this problem. In this paper, we address data-driven solvers for physics, but with additional requirements on the behavior of the simulator:

R1. **Data-driven** – the underlying physics equation is assumed to be completely unknown. This includes the PDE, but also the boundary conditions. The dynamics must be discovered from a finite dataset of trajectories, i.e. a collection of observed behaviors from the physical system,

R2. **Generalization** – the method must be capable of handling new initial conditions that do not explicitly belong to the training set, without re-training or fine-tuning,

R3. **Time and space continuous** – the domain of the predicted solution must be continuous in space and time[1] so that it can be queried at any arbitrary location within the domain of definition.

These requirements are common in the field but rarely addressed altogether. R1 allows for handling complex phenomena where the exact equation might be unknown, and R2 supports the growing need for faster simulators, which consequently must handle new ICs. Space and time continuity (R3) are also useful properties for standard simulations since the solution can be made as fine as needed in certain complex areas.

This task requires learning from sparsely distributed observations only, and without any prior knowledge on the PDE form. In these settings, a standard approach consists of approximating the behavior of a discrete solver, enabling forecasting in an auto-regressive fashion Pfaff et al. (2020); Janny et al. (2023); Sanchez-Gonzalez et al. (2020), losing therefore spatial and temporal continuity. Indeed, auto-regressive models assume strong regularities in the data, such as a static spatial lattice and uniform time steps. For these reasons, generalization to new spatial locations or intermediate time steps is not straightforward. These methods satisfy R1 and R2, but not R3. In another trend, Physics-Informed Neural Networks (PINNs) learn a solution on a continuous domain. They leverage the PDE operator to optimize the weights of a neural network representing the solution, and cannot generalize to new ICs, thus violating R1 and R2.

In this paper, we address R1, R2 and R3 altogether in a new setup involving two joint dynamical systems. R1 and R2 are satisfied using an auto-regressive discrete-time dynamics learned from the sparse observations and producing a trajectory in latent space. Then, R3 is achieved with a state observer derived from a second dynamical system in continuous time. This state observer relies on transformer-based cross-attention to enable evaluation at arbitrary spatio-temporal locations. In a nutshell: **(a)** We propose a new setup to address continuous space and time simulations of physical systems from sparse observation, leveraging insights from control theory. **(b)** We provide strong theoretical results indicating that our setup is well-suited to address this task compared to existing baselines, which are confirmed experimentally on challenging benchmarks. **(c)** We provide experimental evidence that our state observer is more powerful than handcrafted interpolations for the targeted task. **(d)** With experiments on three challenging standard datasets (*Navier* Yin et al. (2022); Stokes (2009), *Shallow Water* Yin et al. (2022); Galewsky et al. (2004), *Eagle* Janny et al. (2023), and against state-of-the-art methods (MeshGraphNet (MGN) Pfaff et al. (2020), DINO Yin et al. (2022), MAgNet (Boussif et al., 2022)), we show that our results generalize to a wider class of problems, with excellent performances.

## 2 RELATED WORKS

**Autoregressive models** – have been extensively used to replicate the behavior of iterative solvers in discrete time, especially in cases where the PDE is unknown or generalization to new initial conditions is needed. These models come in various internal architectures, including convolution-based models for systems observed on a dense uniform grid (Stachenfeld et al., 2021; Guen & Thome, 2020; Bézenac et al., 2019) and graph neural networks (Battaglia et al., 2016) that can adapt to arbitrary spatial discretizations (Sanchez-Gonzalez et al., 2020; Janny et al., 2022a; Li et al., 2018). Such models have demonstrated a remarkable capacity to produce highly accurate predictions and generalize over long prediction horizons, making them particularly suitable for addressing complex problems such as fluid simulation (Pfaff et al., 2020; Han et al., 2021; Janny et al., 2023). However, auto-regressive models are inherently limited to a fixed and constant spatio-temporal discretization grid, hindering their capability to evaluate the solution anywhere and at any time. Neural ordinary differential equations (Neural ODE Chen et al. (2018); Dupont et al. (2019)) offer a countermeasure to the fixed timestep constraint by learning continuous ODEs on discrete data using an explicit solver, such as Euler or Runge-Kutta methods. In theory, this enables the solution to be evaluated at any temporal location but in practice still relies on the discretization of the time variable. Moreover, extending this approach to PDEs is not straightforward. Contrarily to these approaches, we leverage the auto-regressive capacity and accuracy while allowing arbitrary evaluation of the solution at any point in both time and space.

---

[1]In what follows, while being a misnomer, *space and time continuity* of the solution designate the continuity of the spatial and temporal domain of definition of the solution, and not the continuity of the solution itself.

**Continuous solutions for PDEs** – date back to the early days of deep learning (Dissanayake & Phan-Thien, 1994; Lagaris et al., 1998; Psichogios & Ungar, 1992) and have recently experienced a resurgence of interest Raissi et al. (2019; 2017). Physics-informed neural networks represent the solution directly as a neural network and train the model to minimize a residual loss derived from the PDE. They are mesh-free, which alleviates the need for complex adaptive mesh refinement techniques (mandatory in finite volume methods), and have been successfully applied to a broad range of physical problems (Lu et al., 2021; Misyris et al., 2020; Zoboli et al., 2022; Kissas et al., 2020; Yang et al., 2019; Cai et al., 2021), with a growing community proposing architecture designs specifically tailored for PDEs (Sitzmann et al., 2020; Fathony et al., 2021) as well as new training methods (Zeng et al., 2023; Finzi et al., 2023; de Avila Belbute-Peres & Kolter, 2023). Yet, these models are also known to be difficult to train efficiently (Krishnapriyan et al., 2021; Wang et al., 2022). Recently, neural operators have attempted to learn a mapping between function space, leveraging kernels in Fourier space (Li et al., 2020b) (FNO) or graphs (Li et al., 2020a) (GNO) to learn the correspondence from the initial condition to the solution at a fixed horizon. While some operator learning frameworks can theoretically generalize to unseen initial conditions and arbitrary locations, we must consider the practical limitations of existing baselines. For instance, FNO requires a static cartesian grid and cannot be directly evaluated outside the training grid. Similarly, GNO can handle arbitrary meshes in theory, but still has limitations in evaluating points outside the training grid and Li et al. (2021) variant can only be queried at fixed time increments. DeepONet (Lu et al., 2019) can handle free sampling in time and space but is also constrained to a static observation grid.

**Continuous and generalizable solvers** – represent a significant challenge. Few models satisfy all these conditions. MP-PDE (Brandstetter et al., 2022) can handle free-form grids but cannot generalize to different resolutions between train and test, and performs auto-regressive temporal forecasting. Closer to our work, MAgNet (Boussif et al., 2022) proposes to interpolate the observation graph in latent space to new query points before forecasting the solution using graph neural networks. However, they assume prior knowledge of the evaluation mesh and the new query points, use nearest neighbor interpolation instead of trained attention and struggle to generalize to finer grids during test time. In Hua et al. (2022), the auto-regressive MeshGraphNet (Pfaff et al., 2020) is combined with *Orthogonal Spline Collocation* to allow for arbitrary spatial queries. Finally, DINo (Yin et al., 2022) proposes a mesh-free, space-time continuous model to address PDE solving. The model uses context adaptation techniques to dynamically adapt the output of an implicit neural representation forward in time. DINo assumes the existence of a latent ODE modeling the temporal evolution of the context vector and learns it as a Neural ODE. In contrast, our method differs from DINo as our model is based on physics forecasting in an auto-regressive manner. We achieve space and time continuity through a learned dynamical attention transformer capable of handling arbitrary locations and points in time. Our design choices allow for generalization on new spatial and temporal locations, ie. not limited to discrete time steps, and new initial conditions while being trainable from sparse observations [2].

## 3 CONTINUOUS SOLUTIONS FROM SPARSE OBSERVATIONS

Consider a dynamical system following a Partial Differential Equation (PDE) defined for all $(\boldsymbol{x}, t) \in \Omega \times [\![0, T]\!]$, with $T$ a positive constant:

$$
\begin{aligned}
\dot{\boldsymbol{s}}(\boldsymbol{x}, t) &= f(\boldsymbol{s}, \boldsymbol{x}, t) \quad \forall (\boldsymbol{x}, t) \in \Omega \times [\![0, T]\!], \\
\boldsymbol{s}(\boldsymbol{x}, 0) &= \boldsymbol{s}_0(\boldsymbol{x}) \quad \forall \boldsymbol{x} \in \Omega, \quad \boldsymbol{s}(\boldsymbol{x}, t) = \bar{\boldsymbol{s}}(\boldsymbol{x}, t) \quad \forall (\boldsymbol{x}, t) \in \partial\Omega \times [\![0, T]\!]
\end{aligned}
\tag{1}
$$

where the state lies in an invariant set $\boldsymbol{s} \in \mathcal{S}$, $f : \mathcal{S} \mapsto \mathcal{S}$ is an unknown operator, $\boldsymbol{s}_0 : \Omega \mapsto \mathbb{R}^n$ is the initial condition (IC) and $\bar{\boldsymbol{s}} : \partial\Omega \times [\![0, T]\!] \mapsto \mathbb{R}^n$ the boundary condition. In what follows, we consider trajectories with shared boundary conditions, hence we omit $\bar{\boldsymbol{s}}$ from the notation for readability. In practice, the operator $f$ is unknown, and we assume access to a set $\mathcal{D}$ of $K$ discrete trajectories from different ICs, $\boldsymbol{s}_0^k$, sampled at sparse and scattered locations in time and space. Formally, we introduce two finite sets $\mathcal{X} \subset \Omega$ of fixed positions and fixed regularly sampled times $\mathcal{T}$ at sampling rate $\Delta^*$. Let $S(\boldsymbol{s}_0, \boldsymbol{x}, t)$ be the solution of this PDE from IC $\boldsymbol{s}_0$, the dataset $\mathcal{D}$ is given as: $\mathcal{D} := \left\{ S(\boldsymbol{s}_0^k, \mathcal{X}, \mathcal{T}) \,\middle|\, k \in [\![1, K]\!] \right\}$. Our task is formulated as:

*Given $\mathcal{D}$, a new initial condition $\boldsymbol{s}_0 \in \mathcal{S}$, and a query $(\boldsymbol{x}, t) \in \Omega \times [\![0, T]\!]$, find the solution of equation 1 at the queried location and from the given IC, that is $S(\boldsymbol{s}_0, \boldsymbol{x}, t)$.*

---

[2]Code will be made public. Project page: https://continuous-pde.github.io/

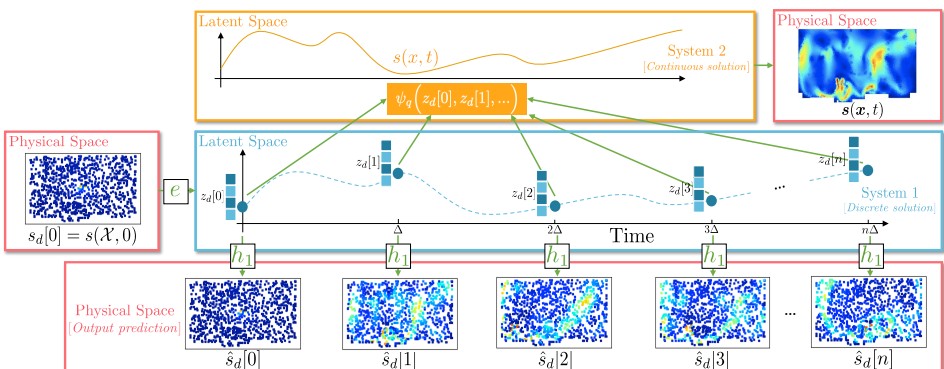

Figure 1: **Model overview** – We achieve space and time continuous simulations of physics systems by formulating the task as a double observation problem. System 1 is a discrete dynamical model used to compute a sequence of latent anchor states $\boldsymbol{z}_d$ auto-regressively, and System 2 is used to design a state estimator $\psi_q$ retrieving the dense physical state at arbitrary locations $(\boldsymbol{x}, t)$.

Note that this task involves generalization to new ICs, as well as estimation to unseen spatial locations within $\Omega$ and unseen time instants within $[\![0, T]\!]$. We do not explicitly require extrapolation to instants $t > T$, although it comes as a side benefit of our approach up to some extent.

### 3.1 THE DOUBLE OBSERVATION PROBLEM

The task implies extracting regularities from weakly informative physical variables that are sparsely measured in space and time, since $\mathcal{X}$ and $\mathcal{T}$ contain very few elements. Consequently, the possibility to forecast their trajectories from off-the-shelf auto-regressive methods is very unlikely (as confirmed experimentally). To tackle this challenge, we propose an approach accounting for the fact that the phenomenon is not directly observable from the sparse trajectories, but can be deduced from a richer latent state-space in which the dynamics is markovian. We introduce two linked dynamical models lifting sparse observations to dense trajectories guided by *observability* considerations, namely

$$\underline{\text{System 1:}} \left\{ \begin{array}{ll} \boldsymbol{z}_d[n+1] & = f_1\big(\boldsymbol{z}_d[n]\big) \\ \boldsymbol{s}_d[n] & = h_1\big(\boldsymbol{z}_d[n]\big) \end{array} \right. , \quad \underline{\text{System 2:}} \left\{ \begin{array}{ll} \dot{\boldsymbol{s}}(\boldsymbol{x}, t) & = f_2\big(\boldsymbol{s}, \boldsymbol{x}, t\big) \\ \boldsymbol{z}(\boldsymbol{x}, t) & = h_2\big(\boldsymbol{s}, \boldsymbol{x}, t\big) \end{array} \right. \quad \forall (\boldsymbol{x}, t) \in \Omega \times [\![0, T]\!] \tag{2}$$

where for all $n \in \mathbb{N}$, we note $\boldsymbol{s}_d[n] = \boldsymbol{s}(\mathcal{X}, n\Delta)$ the sparse observation at some instant $n\Delta$ (the sampling rate $\Delta$ is not necessarily equal to the sampling rate $\Delta^*$ used for data acquisition, which we will exploit during training to improve generalization. This will be detailed later).

**System 1** – is a discrete-time dynamical system where the available measurements $\boldsymbol{s}_d[n]$ are considered as *partial observations* of a latent state variable $\boldsymbol{z}_d[n]$. We aim to derive an output predictor from System 1 to forecast trajectories of sparse observations auto-regressively from the sparse IC. As mentioned earlier, sparse observations are unlikely to be sufficient to perform predictions, hence we introduce a richer latent state variable $\boldsymbol{z}_d$ in which the dynamics is truly markovian, and observations $\boldsymbol{s}_d[n]$ are seen as measurements of the state $\boldsymbol{z}_d$ using the function $h_1$.

**System 2** – is a continuous-time dynamical system describing the evolution of the to-be-predicted dense trajectory $S(\boldsymbol{s}_0, \boldsymbol{x}, t)$. It introduces continuous observations $\boldsymbol{z}(\boldsymbol{x}, t)$ such that $\boldsymbol{z}(\mathcal{X}, n\Delta) = \boldsymbol{z}_d[n]$. The insight is that the state representation $\boldsymbol{z}_d[n]$ obtained from System 1 is designed to contain sufficient information to predict $\boldsymbol{s}_d[n]$, but not necessarily to predict the dense state. Formally, $\boldsymbol{z}_d$ represents solely the *observable part* of the state, in the sense of control theory.

At inference time, we forecast at query location $(\boldsymbol{x}, t)$ with a 2-step algorithm: **(Step-1)** System 1 is used as an output predictor from the sparse IC $\boldsymbol{s}_d[0]$, and computes a sequence $\boldsymbol{z}[0], \boldsymbol{z}[1], ...,$ which we refer to as "*anchor states*". This sequence allows the dynamics to be Markovian, provides sufficient information for the second state estimation step and holds information to predict the sparse observations, allowing supervision during training. **(Step-2)** We derive a *state observer* from System 2 leveraging the anchor states over the whole time domain to estimate the dense solution at an arbitrary location in space and time (see figure 1). Importantly, for a given IC, the anchor states are computed only once and reused within System 2 to estimate the solution at different points.

## 3.2 THEORETICAL ANALYSIS

In this section, we introduce theoretical results supporting the use of Systems 1 and 2. In particular, we show that using System 1 to forecast the sparse observations in latent space $z_d$ rather than directly operating in the physical space leads to smaller upper bounds on the prediction error. Then, we show the existence of a state estimator from System 2 and compute an upper bound on the estimation error depending on the length of the sequence of anchor states.

**Step 1** – consists of computing the sequence of anchor states guided by an output prediction task of the sparse observations. As classically done, we introduce an encoder (formally, a state observer) $e(s_d[0]) = z_d[0]$ coupled to System 1 to project the sparse IC into a latent space $z_d$. Following System 1, we compute the anchor states $z_d$ auto-regressively (with $f_1$) in the latent space. The sparse observations are extracted from $z_d$ using $h_1$. In comparison, existing baselines (Pfaff et al., 2020; Sanchez-Gonzalez et al., 2020; Stachenfeld et al., 2021) maintain the state in the physical space and discard the intermediate latent representation between iterations. Formally, let us consider approximations $\hat{f}_1, \hat{h}_1, \hat{e}$ (in practice realized as deep networks trained from data $\mathcal{D}$) of $f_1, h_1$ and $e$ and compare the prediction algorithm for the classic auto-regressive (AR) approach and ours

$$\text{Classic AR:} \quad \hat{s}_d^{\text{ar}}[n] := (\hat{h}_1 \circ \hat{f}_1 \circ \hat{e})^n (s_d[0]) \qquad \text{Ours:} \quad \hat{s}_d[n] := \hat{h}_1 \circ \hat{f}_1^n \circ \hat{e}(s_d[0]) \qquad (3)$$

Classical AR approaches re-project the latent state into the physical space at each step and repeat "encode-process-decode". Our method encodes the sparse IC, advances the system in the latent space, and decodes toward the physical space at the end. A similar approach has also been explored in Wu et al. (2022); Kochkov et al. (2020), albeit in different contexts, without theoretical analysis.

**Proposition 1** *Consider a dynamical system of the form of System 1 and assume the existence of a state observer $e$ along with approximations $\hat{f}_1, \hat{h}_1, \hat{e}$ with Lipschitz constants $L_f, L_h$ and $L_e$ respectively such that $L_h L_f L_e \neq 1$. If there exist $\delta_f, \delta_h, \delta_e \in \mathbb{R}^+$ such that $\forall (z, s) \in \mathbb{R}^{n_z} \times \mathbb{R}^{n_s}$*

$$|f_1(z) - \hat{f}_1(z)| \leqslant \delta_f, \quad |h_1(z) - \hat{h}_1(z)| \leqslant \delta_h, \quad |e(s) - \hat{e}(s)| \leqslant \delta_e \qquad (4)$$

*for the Euclidean norm $|\cdot|$, then for all integer $n > 0$, with $\hat{s}_d[n]$ and $\hat{s}_d^{\text{ar}}[n]$ as in equation 3,*

$$|s_d[n] - \hat{s}_d[n]| \leqslant \delta_h + L_h \left( \delta_f \frac{L_f^n - 1}{L_f - 1} + L_f^n \delta_e \right) \qquad (5)$$

$$|s_d[n] - \hat{s}_d^{\text{ar}}[n]| \leqslant \delta \frac{L^n - 1}{L - 1} \qquad (6)$$

*with $\delta = \delta_h + L_h \delta_f + L_h L_f \delta_e$ and $L = L_h L_f L_e$.*

*Proof*: See appendix B.

This result shows that falling back to the physical space at each time step degrades the upper bound of the prediction error. Indeed, if $L < 1$, the upper bound converges trivially to zero when $n$ increases, and hence can be ignored. Otherwise, the upper bound for the classic AR scheme appears to be more sensitive to approximation errors $\delta_h, \delta_f$ and $\delta_e$ compared to our approach (for a formal comparison, see appendix C). Intuitively it means that information is lost in the observation space, which thus needs to be re-estimated at each iteration when using the classic AR scheme. By maintaining a state variable in the latent space, we allow this information to flow readily between each step of the simulator (see blue frame in figure 1).

**Step 2** – The state estimator builds upon System 2 and relies on the set of anchor states from the previous step to estimate the dense physical state at arbitrary locations in space and time. Formally, we look for a function $\psi_q$ leveraging the sequence of anchor states $z_d[0], \cdots z_d[q]$ (simulated from the sparse IC $s_d[0]$) to retrieve the dense solution[3]. In what follows, we show that (1) such a function $\psi_q$ exists and (2) we compute an upper bound on the estimation error depending on the length of the sequence. To do so, consider the functional which outputs the anchor states from any IC $s_0 \in \mathcal{S}$

$$\mathcal{O}_p(s_0) = \left[ \ h_2(s_0(\mathcal{X})) \ h_2(S(s_0, \mathcal{X}, \Delta)) \ \cdots \ h_2(S(s_0, \mathcal{X}, p\Delta)) \ \right] = \left[ \ z_d[0] \ \cdots \ z_d[p] \ \right] \qquad (7)$$

In practice, the ground truths $z_d[n]$ are not perfectly known, as they are obtained from a data-driven output predictor (step 1) using the sparse IC. Inspired from Janny et al. (2022b), we state:

---

[3]Since the simulation is conducted up to $T$, and considering the time step $\Delta$, in practice $q \leqslant \lfloor \frac{T}{\Delta} \rfloor$

**Proposition 2** *Consider a dynamical system defined by System 2 and equation 7. Assume that*

*A1.* $f_2$ *is Lipschitz with constant $L_s$,*
*A2. there exists $p > 0$ and a strictly increasing function $\alpha$ such that $\forall \boldsymbol{s}_a, \boldsymbol{s}_b \in \mathcal{S}^2$ and $\forall q \geqslant p$*

$$\left| \mathcal{O}_q(\boldsymbol{s}_a) - \mathcal{O}_q(\boldsymbol{s}_b) \right| \geqslant \alpha(q) |\boldsymbol{s}_a - \boldsymbol{s}_b|_{\mathcal{S}} \tag{8}$$

*where $\left| \cdot \right|_{\mathcal{S}}$ is an appropriate norm for $\mathcal{S}$.*

*Then, $\forall q \geqslant p$, there exists $\psi_q$ such that, for $(\boldsymbol{x}, t) \in \Omega \times [\![0, T]\!]$ and $\delta_n$ such that $\hat{\boldsymbol{z}}_d[n] = \boldsymbol{z}_d[n] + \delta_n$, for all $n \leqslant q$,*

$$\psi_q\big(\boldsymbol{z}_d[0], \cdots, \boldsymbol{z}_d[q], \boldsymbol{x}, t\big) = S(\boldsymbol{s}_0, \boldsymbol{x}, t) \tag{9}$$

$$\left| S(\boldsymbol{s}_0, \boldsymbol{x}, t) - \psi_q\big(\hat{\boldsymbol{z}}_d[0], \cdots, \boldsymbol{z}_d[q], \boldsymbol{x}, t\big) \right|_{\mathcal{S}} \leqslant 2\alpha(q)^{-1} \big| \delta_{0|q} \big| e^{L_s t}. \tag{10}$$

*where $\delta_{0|q} = \begin{bmatrix} \delta_0 & \cdots & \delta_q \end{bmatrix}$.*

*Proof:* See appendix D. Assumption A2. states that the longer we observe two trajectories from different ICs, the easier it will be to distinguish them, ruling out systems collapsing to the same state. Such systems are uncommon since forecasting their trajectory becomes trivial after some time. This assumption is related to *finite-horizon observability* in control theory, a property of dynamical systems guaranteeing that the (markovian) state can be retrieved given a finite number $p$ of past observations. Equation 8 is associated with injectivity of $\mathcal{O}_q$, hence the existence of a left inverse mapping the sequence of anchor states to the IC $\boldsymbol{s}_0$.

Proposition 2 highlights a trade-off on the performance of $\psi_q$. On one hand, longer sequences of anchor states are harder to predict, leading to a larger $|\delta_{0|q}|$, which impacts the state estimator $\psi_q$ negatively. On the other hand, longer sequences hold more information that can still be leveraged by $\psi_q$ to improve its estimation, represented by $\alpha(q)^{-1}$ in equation 10. In contrast to competing baselines or conventional interpolation algorithms, our approach takes this trade-off into account, by explicitly leveraging the sequence to estimate the dense solution, as will be discussed below.

**Discussion and related work** – the competing baselines can be analyzed using our setup, yet in a weaker configuration. For instance, one can see Step 2 as an interpolation process, and replace it with a conventional interpolation algorithm, which typically relies on spatial neighbors only. Our method not only exploits spatial neighborhoods but also leverages temporal data, improving the performance, as shown in proposition 2 and empirically corroborated in Section 4.

MAgNet (Boussif et al., 2022) uses a reversed *interpolate-forecast* scheme compared to ours. The IC $\boldsymbol{s}_d[0]$ is interpolated right from the start to estimate $\boldsymbol{s}_0$ (corresponding to our Step 2, with $q{=}1$), and then simulated with an auto-regressive model in the physical space (with the classic AR scheme). Propositions 1 and 2 show that the upper bounds on the estimation and prediction error are higher than ours. Moreover, if the number of query points exceeds the number of known points ($|\Omega| \gg |\mathcal{X}|$), the input of the auto-regressive solver is filled with noisy interpolations, which impacts performance.

DINo (Yin et al., 2022) is a very different approach leveraging a spatial implicit neural representation modulated by a context vector, whose dynamics is modeled via a learned ODE. This approach is radically different than ours and arguably involves stronger hypotheses, such as the existence of a learnable ODE modeling the dynamics of a suitable weight modulation vector. In contrast, our method relies on arguably more sound assumptions, i.e. the existence of an observable discrete dynamics explaining the sparse observation, and the finite-time observability of System 2.

### 3.3 IMPLEMENTATION

The implementation follows the algorithm described in the previous section: **(Step-1)** rolls out predictions of anchor states from the IC, **(Step-2)** estimates the state at the query position from these anchor states. The encoder $\hat{e}$ from Step 1 is a multi-layer perceptron (MLP) which takes as input the sparse IC $\boldsymbol{s}_d[0]$ and the positions $\mathcal{X}$ and outputs a latent state variable $\boldsymbol{z}_d[0]$ structured as a graph, with edges computed with a Delaunay triangulation. Hence, each anchor is a graph $\boldsymbol{z}_d[n] = \{\boldsymbol{z}_d[n]_i\}$, but we will omit index $i$ over graph nodes in what follows if not required for understanding.

We model $\hat{f}_1$ as a multi-layer Graph Neural Network (GNN) (Battaglia et al., 2016). The anchor states $\boldsymbol{z}_d[n]$ are defined at fixed time steps $n\Delta$, which might not match $\Delta^*$ used in the data $\mathcal{T}$. We

found it beneficial to choose $\Delta=k\times\Delta^*$ with $k>1\in\mathbb{N}$ such that the model can be queried *during training* on time points $t\in\mathcal{T}$ that do not match exactly with every time-steps in $\boldsymbol{z}_d[0],\boldsymbol{z}_d[1],...,$ but rather on a subset of them, hence encouraging generalization to unseen time. The observation function $\hat{h}_1$ is an MLP applied on the vector at node level in the graph $\boldsymbol{z}_d$.

The state estimator $\psi_q$ is decomposed into a Transformer model (Vaswani et al., 2017) coupled to a recurrent neural network to provide an estimate at query spatio-temporal query position $(\boldsymbol{x},t)$. First, through cross-attention we translate the set of anchor states $\boldsymbol{z}_d[n]$ (one embedding per graph node $i$ and per instant $n$) into a set of estimates of the continuous variable $\boldsymbol{z}(\boldsymbol{x},t)$ *conditioned* at the instant $n\Delta$, which we denote $\boldsymbol{z}_{n\Delta}(\boldsymbol{x},t)$ (one embedding per instant $n$). Following advances in geometric mappings in computer vision (Saha et al., 2022), we use multi-head cross-attention to *query* from coordinates $(\boldsymbol{x},t)$ to *Keys* corresponding to the nodes $i$ in each graph anchor state $\boldsymbol{z}_d[n],\forall n$:

$$\boldsymbol{z}_{n\Delta}(\boldsymbol{x},t)=f_{\mathrm{mha}}\big(\mathrm{Q}{=}\zeta_\omega(\boldsymbol{x},t),\mathrm{K}{=}\mathrm{V}{=}\{\boldsymbol{z}_d[n]_i\}+\zeta_\omega(\mathcal{X},n\Delta)\big),\texttt{ // attention over nodes i} \quad (11)$$

where $Q,K,V$ are, respectively, *Query*, *Key* and *Value* inputs to the cross-attention layer $f_{\mathrm{mha}}$ (Vaswani et al., 2017) and $\zeta_\omega$ a Fourier positional encoding with a learned frequency parameter $\omega$. Finally, we leverage a state observer to estimate the dense solution at the query point from the sequence of conditioned anchor variables, over time. This is achieved with a Gated Recurrent Unit (GRU) Cho et al. (2014) maintaining a hidden state $\boldsymbol{u}[n]$,

$$\boldsymbol{u}[n]=r_{\mathrm{gru}}\big(\boldsymbol{u}[n{-}1],\boldsymbol{z}_{n\Delta}(\boldsymbol{x},t)\big),\quad\hat{S}(\boldsymbol{s}_0,\boldsymbol{x},t)=D\left(\boldsymbol{u}[q]\right), \quad (12)$$

which shares similarities with conventional state-observer designs in control theory (Bernard et al., 2022). Finally, an MLP $D$ maps the final GRU hidden state to the desired output, that is, the value of the solution at the desired spatio-temporal coordinate $(\boldsymbol{x},t)$. See appendix E for details.

## 3.4 TRAINING

Generalization to new input locations during training is promoted by creating artificial generalization situations using sub-sampling techniques of the sparse sets $\mathcal{X}$ and $\mathcal{T}$.

**Artificial generalization** – The anchor states $z_d[n]$ are computed at time rate $\Delta$ larger than the available rate $\Delta^*$. This creates situations *during training* where the state estimator $\psi_q$ does not have access to a latent state perfectly matching with the queried time. We propose a similar trick to promote spatial generalization. At each iteration, we sub-sample the (already sparse) IC $\boldsymbol{s}_d[0]$ randomly to obtain $\tilde{\boldsymbol{s}}_d[0]$ defined on a subset of $\mathcal{X}$. We then compute the anchor states $\tilde{\boldsymbol{z}}_d$ using System 1. On the other hand, the query points are selected in the larger set $\mathcal{X}$. Consequently, System 2 is exposed to positions that do not always match with the ones in $\boldsymbol{z}_d[n]$. Note that the complete domain of definition $\Omega\times[\![0,T]\!]$ remains unseen during training.

**Training objective** – To reduce training time, we randomly sample $M$ query points $(\boldsymbol{x}_m,\tau_m)$ in $\mathcal{X}\times\mathcal{T}$ at each iteration, with a probability proportional to the previous error of the model at this point since its last selection (see appendix E) and we minimize the loss

$$\mathcal{L}=\sum_{k=1}^{K}\overbrace{\sum_{m=1}^{M}\left|S(\boldsymbol{s}_0^k,\boldsymbol{x}_m,\tau_m)-\psi_q\big(\tilde{\boldsymbol{z}}_d[0|q],\boldsymbol{x},\tau_m\big)\right|^2}^{\mathcal{L}_{\mathrm{continuous}}}+\overbrace{\sum_{n=0}^{\lfloor T/\Delta\rfloor}\left|\tilde{\boldsymbol{s}}_d[n]-\hat{h}_1\big(\tilde{\boldsymbol{z}}_d[n]\big)\right|^2}^{\mathcal{L}_{\mathrm{dynamics}}}, \quad (13)$$

with $\tilde{\boldsymbol{z}}_d[n]=\hat{f}_1^{\,n}\circ\hat{e}\big(\tilde{\boldsymbol{s}}_d[0]\big)$. $\mathcal{L}_{\mathrm{continuous}}$ supervises the model end-to-end, and $\mathcal{L}_{\mathrm{dynamics}}$ trains the latent anchor states $\boldsymbol{z}_d$ to predict the sparse observations from the IC.

## 4 EXPERIMENTAL RESULTS

**Experimental setup** – $\mathcal{X}\times\mathcal{T}$ results from sub-sampling $\Omega\times[\![0,T]\!]$ with different rates to control the difficulty of the task. We evaluate on three highly challenging datasets (details in appendix F): **Navier** (Yin et al., 2022; Stokes, 2009) simulates the vorticity of a viscous, incompressible flow driven by a sinusoidal force acting on a square domain with periodic boundary conditions. **Shallow Water** (Yin et al., 2022; Galewsky et al., 2004) studies the velocity of shallow waters evolving on the tangent surface of a 3D sphere. **Eagle** (Janny et al., 2023) is a challenging dataset of turbulent airflow generated by a moving drone in a 2D environment with many different scene geometries.

We evaluate our model against three baselines representing the state-of-the-art in continuous simulations. **Interpolated MeshGraphNet (MGN)** (Pfaff et al., 2020) is a standard multi-layered GNN

| | | Navier | | | Shallow Water | | | Eagle | |
|---|---|---|---|---|---|---|---|---|---|
| | | High | Mid | Low | High | Mid | Low | High | Low |
| DINo | In-$\mathcal{X}$ | 1.557 | 1.130 | 1.878 | **0.1750** | **0.1814** | 0.2733 | 287.3 | 302.7 |
| (Yin et al., 2022) | Ext-$\mathcal{X}$ | 1.600 | 1.253 | 5.493 | 4.638 | 13.40 | 21.55 | 381.7 | 489.6 |
| Interp. MGN | In-$\mathcal{X}$ | 1.913 | 0.9969 | 0.6012 | 0.3663 | 0.2835 | 0.7309 | **64.44** | 83.58 |
| (Pfaff et al., 2020) | Ext-$\mathcal{X}$ | 2.694 | 4.784 | 14.80 | 1.744 | 4.221 | 8.187 | 173.4 | 241.5 |
| *Time Oracle (n.c)* | In-$\mathcal{X}$ | | *n/a* | | | *n/a* | | | *n/a* | |
| | Ext-$\mathcal{X}$ | *0.851* | *4.204* | *15.63* | *1.617* | *4.327* | *8.522* | *147.0* | *221.2* |
| MAgNet | In-$\mathcal{X}$ | 18.17 | 6.047 | 8.679 | 0.3196 | 0.3358 | 0.4292 | 99.79 | 124.5 |
| (Boussif et al., 2022) | Ext-$\mathcal{X}$ | 35.73 | 26.24 | 57.21 | 10.21 | 23.20 | 30.55 | 194.3 | 260.7 |
| **Ours** | In-$\mathcal{X}$ | **0.1989** | **0.2136** | **0.2446** | 0.2940 | 0.3139 | **0.2700** | 70.02 | **78.83** |
| | Ext-$\mathcal{X}$ | **0.2029** | **0.2463** | **0.5601** | **0.4493** | **1.051** | **2.800** | **90.88** | **117.2** |

Table 1: **Space Continuity** – we evaluate the spatial interpolation power of our method vs. the baselines and standard interpolation techniques. We vary the number of available measurement points in the data for training from High (25% of simulation grid), Middle (10%), and Low (5%) amount of points and show that our model outperforms the baselines. Evaluation is conducted over 20 frames in the future (10 for *Eagle*) and we report the MSE to the ground truth solution ($\times 10^{-3}$).

used auto-regressively and extended to spatiotemporal continuity using physics-agnostic interpolation. **MAgNet** (Boussif et al., 2022) interpolates the IC at the query position in latent space before using MGN. The original implementation assumes knowledge of the target graph during training, including new queries. When used for superresolution, the authors kept the ratio between the amount of new query points and available points constant. Hence, while MAgNet is queried at unseen locations, it also benefits from more information. In our setup, the model is exposed to a fixed number of points but does not receive more samples during evaluation. This makes our problem more challenging than the one addressed in Boussif et al. (2022). **DINo** (Yin et al., 2022) models the solution as an Implicit Neural Representation (INR) $s(x, \alpha_t)$ where the spatial coordinates $x$ are fed to a MFN (Fathony et al., 2021) and $\alpha_t$ is a context vector modulating the weights of the INR. The dynamics of $\alpha$ is modeled with a Neural-ODE, where the dynamics is an MLP. We share common objectives with DINo and take inspiration from their evaluation tasks yet in a more challenging setup. Details of the baselines are in appendix F. We highlight a caveat on MAgNet: the model can handle a limited amount of new queries, roughly equal to the number of observed points. Our task requires the solution at up to 20 times more queries than available points. In this situation, the graph in MaGNet is dominated by noisy states from interpolation, and the auto-regressive forecaster performs poorly. During evaluation, we found it beneficial to split the queries into chunks of 10 nodes and to apply the model several times. This strongly improves the performance at the cost of an increased runtime.

**Space Continuity** – Table 1 compares the spatial interpolation power of our method versus several baselines. The MSE values computed on the training domain (In-$\mathcal{X}=\mathcal{X}$) and outside (Ext-$\mathcal{X}=\Omega \setminus \mathcal{X}$) show that our method offers the best performance, especially for the Ext-domain task, which is our aim. To ablate dynamics and evaluate the impact of trained interpolations, we also report the predictions of a *Time Oracle* which uses sparse ground truth values at all time steps and interpolates (bicubic) spatially. This allows us to assess whether the method is doing better than a simple axiomatic interpolation. While MGN offers competitive in-domain predictions, the cubic interpolation fails to extrapolate reliably on unseen points. This can be seen in the In/Ext gap for Interpolated MGN which is very close to the Time Oracle error. MaGNet, which builds on a similar framework, is hindered by the larger amount of unobserved data in the input mesh. At test time, the same number of initial condition points are provided but the method interpolates substantially more points. DINo achieves a very low In/Ext gap, yet fails on highly (5%) down-sampled tasks. One of the key difference with DINo is that the dynamics relies on an internal ODE for the temporal evolution of a modulation vector. In contrast, our model uses an explicit auto-regressive backbone, and time forecasting is handled in an arguably more meaningful space, which we conjecture to be the reason why we achieve better results (see fig. 5 in the appendix).

**Time Continuity** – is a step forward in difficulty, as the model needs to interpolate not only to unseen spatial locations (datasets are undersampled at 25%) but also on intermediate timesteps (Ext-$\mathcal{T}$, Table 2). All models perform well on *Shallow Water*, which is relatively easy. Both DINo and MAgNet leverage a discrete integration scheme (Euler for MAgNet and RK4 for DINo) allowing querying the model between timesteps seen at training. These schemes struggle to capture the data dependencies effectively and therefore the methods fail on *Navier* (see also Figure 6 for qualitative

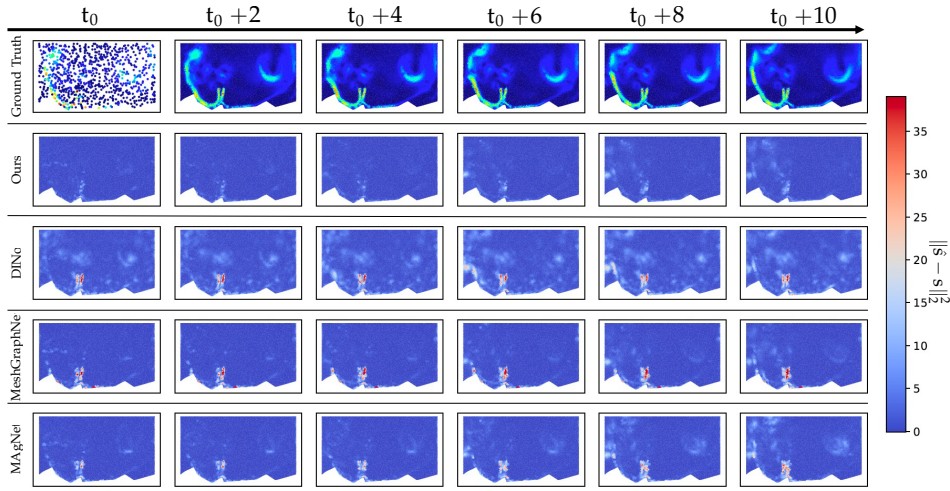

Figure 2: **Results on Eagle** – Per point error of the flow prediction on an *Eagle* example in the *Low* spatial down-sampling scenario. Our model exhibits lower errors as also shown in Tables 1 and 2.

| | | Navier | | | Shallow Water | | | Eagle | | |
|---|---|---|---|---|---|---|---|---|---|---|
| | | 1/1 | 1/2 | 1/4 | 1/1 | 1/2 | 1/4 | 1/1 | 1/2 | 1/4 |
| DINo | In-$\mathcal{T}$ | 1.590 | 36.31 | 46.02 | 3.551 | 6.005 | 6.249 | 444.5 | 447.1 | 448.6 |
| (Yin et al., 2022) | Ext-$\mathcal{T}$ | n/a | 39.42 | 54.72 | n/a | 6.015 | 6.265 | n/a | 479.4 | 470.7 |
| Interp. MGN | In-$\mathcal{T}$ | 2.506 | 4.834 | 12.77 | 1.408 | 1.289 | 1.333 | 203.4 | 210.4 | 263.3 |
| (Pfaff et al., 2020) | Ext-$\mathcal{T}$ | n/a | 5.922 | 36.43 | n/a | 1.287 | 1.355 | n/a | 209.8 | 263.8 |
| *Spatial Oracle (n.c)* | In-$\mathcal{T}$ | | *n/a* | | | *n/a* | | | *n/a* | |
| | Ext-$\mathcal{T}$ | *n/a* | *1.296* | *28.58* | *n/a* | *0.003* | *0.119* | *n/a* | *29.46* | *54.53* |
| MAgNet | In-$\mathcal{T}$ | 31.51 | 135.0 | 243.9 | 7.804 | 6.433 | 1.884 | 227.9 | 220.3 | **225.8** |
| (Boussif et al., 2022) | Ext-$\mathcal{T}$ | n/a | 142.8 | 255.5 | n/a | 6.291 | 1.947 | n/a | 229.8 | **230.6** |
| **Ours** | In-$\mathcal{T}$ | **0.2019** | **0.1964** | **0.4062** | **0.4115** | **0.4278** | **0.4549** | **108.0** | **106.1** | 278.6 |
| | Ext-$\mathcal{T}$ | n/a | **0.2138** | **11.36** | n/a | **0.4326** | **0.4802** | n/a | **119.9** | 306.9 |

Table 2: **Time Continuity** – we evaluate the time interpolation power of our method vs. the baselines. Models are trained and evaluated with 25% of $\Omega$, and with different temporal resolutions (full, half, and quarter of the original). The Spatial Oracle (*not comparable!*) uses the exact solution at every point in space, and performs temporal interpolation. Evaluation is conducted over 20 frames in the future (10 for *Eagle*) and we report MSE compared to the ground truth solution ($\times 10^{-3}$).

results). *Eagle* is particularly challenging, the main source of error being the spatial interpolation, as can be seen in Figure 2 – our method yields lower errors in flow estimation.

**Many more experiments** – are available in appendix G. We study the **impact of key design choices**, artificial generalization, and dynamical loss. We show qualitative results on time interpolation, time extrapolation on the *Navier* dataset. We **explore generalization to different grids**. We provide more empirical evidence of the soundness of Step 2 in an **ablation study** (including comparison with attentive neural process Kim et al. (2018), an attention-based structure somehow close to ours), and **observe attention maps** on several examples. We show that our **state estimator goes beyond local interpolation**, as conventional interpolation algorithms would do. Finally, we also measure the **computational burden** of the discussed methods and show that our approach is more efficient. 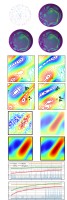

## 5 CONCLUSION

We exploit a double dynamical system formulation for simulating physical phenomena at arbitrary locations in time and space. Our approach comes with theoretical guarantees on existence and accuracy without knowledge of the underlying PDE. Furthermore, our method generalizes to unseen initial conditions and reaches excellent performances outperforming existing methods. Potential applications of our model goes beyond fluid dynamics and can be applied to various PDE-based problem. Yet, our approach relies on several hypotheses such as regular time sampling and observability. Finally, for known and well-studied phenomena, it would be interesting to add physics priors in the system, a nontrivial extension that we leave for future work.

**Reproducibility** – the detailed model architecture is described in the appendix. For the sake of reproducibility, in the case of acceptance, we will provide the source code for training and evaluating our model, as well as trained model weights. For training, we will provide instructions for setting up the codebase, including installing external dependencies, pre-trained models, and pre-selected hyperparameter configuration. For the evaluation, the code will include evaluation metrics directly comparable to the paper's results.

**Ethics statement** – While our simulation tool is unlikely to yield unethical results, we are mindful of potential negative applications of improving fluid dynamics simulations, particularly in military contexts. Additionally, we strive to minimizing the carbon footprint associated with our training processes.

## 6 ACKNOWLEDGEMENTS

We recognize support through French grants "*Delicio*" (ANR-19-CE23-0006) of call CE23 "*Intelligence Artificielle*" and "*Remember*" (ANR-20-CHIA0018), of call "*Chaires IA hors centres*". This work was performed using HPC resources from GENCI- IDRIS (Grant 2023-AD010614014).

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

## A  WEBSITE AND INTERACTIVE ONLINE VISUALIZATION

An anonymous website has been created where results can be visualized with an online interactive tool, which allows one to choose time steps interactively with the mouse, and in the case of the *Shallow Water* dataset, also the orientation of the spherical data:

**https://continuous-pde.github.io/**

## B  PROOF OF PROPOSITION 1

The proof proceeds by successive majorations and triangular inequalities. For sake of clarity, and only in this proof we omit the $d$ subscript and write $s[n]$ and $z[n]$ for $s_d[n]$ and $z_d[n]$, respectively.

We start with $\hat{s}[n] := \hat{h}_1 \circ \hat{f}_1{}^n \circ \hat{e}(s[0])$. Thus for any integer $n > 0$ we have

$$|s[n] - \hat{s}[n]| = |h_1(z[n]) - \hat{h}_1(\hat{z}[n])|. \tag{14}$$

Using Lipschitz property and 4, then

$$|s[n] - \hat{s}[n]| \leqslant |h_1(z[n]) - \hat{h}_1(z[n])| + |\hat{h}_1(z[n]) - \hat{h}_1(\hat{z}[n])| \tag{15}$$
$$\leqslant \delta_h + L_h|z[n] - \hat{z}[n]|.$$

Noticing that one can rewrite $\hat{z}[n]$ as $\hat{z}[n] = \hat{f}_1{}^n \circ \hat{e}(s[0])$. Since $z[n] = f_1{}^n(z[0])$ and using a similar decomposition as for 15), one gets:

$$|z[n] - \hat{z}[n]| \leqslant \delta_f \sum_{k=0}^{n-1} L_f^k + L_f^n |z[0] - \hat{z}[0]|. \tag{16}$$

Hence, from equation 15, and using $z[0] = e(s[0])$ and $\hat{z}[0] = \hat{e}(s[0])$, we have

$$|s[n] - \hat{s}[n]| \leqslant \delta_h + L_h\left(\delta_f \frac{L_f^n - 1}{L_f - 1} + L_f^n \delta_e\right). \tag{17}$$

We now move on to the classic auto-regressive case, i.e. $\hat{s}^{\text{ar}}[n] = \left(\hat{h}_1 \circ \hat{f}_1 \circ \hat{e}\right)^n(s[0])$.

$$|s[n] - \hat{s}^{\text{ar}}[n]| \leqslant \left|h_1(z[n]) - \hat{h}_1(z[n])\right| + \left|\hat{h}_1(z[n]) - \hat{h}_1(\hat{z}^{\text{ar}}[n])\right| \tag{18}$$
$$\leqslant \delta_h + L_h\left|z[n] - \hat{z}^{\text{ar}}[n]\right|$$
$$\leqslant \delta_h + L_h\left(\delta_f + L_f\left|e(s[n-1]) - \hat{e}(\hat{s}^{\text{ar}}[n-1])\right|\right)$$
$$\leqslant \delta_h + L_h\left(\delta_f + L_f\left(\delta_e + L_e|s[n-1] - \hat{s}^{\text{ar}}[n-1]|\right)\right)$$
$$\leqslant \delta \sum_{i=0}^{n-2} L^i + L^{n-1}\left|s[1] - \hat{s}^{\text{ar}}[1]\right|,$$

with $\delta = \delta_h + L_h\delta_f + L_hL_f\delta_e$ and $L = L_hL_fL_e$. Moreover,

$$|s[1] - \hat{s}^{\text{ar}}[1]| = |\hat{h}_1(z[1]) - \hat{f}_1(\hat{z}^{\text{ar}}[1])| \tag{19}$$
$$\leqslant \delta_h + L_h|z[1] - \hat{z}^{\text{ar}}[1]| \tag{20}$$
$$\leqslant \delta_h + L_h(\delta_f + L_f|z[0] - \hat{z}^{\text{ar}}[0]|) \tag{21}$$
$$\leqslant \delta \tag{22}$$

Putting it all together, we get equation 6:

$$|s[n] - \hat{s}^{\text{ar}}[n]| \leqslant \delta\frac{L^n - 1}{L - 1} \tag{23}$$

Finally, equation 17 and equation 23 conclude the proof.

## C   COMPARISON OF UPPER BOUNDS IN PROPOSITION 1

We start by formulating equation 5 and equation 6 under a comparable form

$$|\boldsymbol{s}_d[n] - \hat{\boldsymbol{s}}_d[n]| \leqslant \delta + L_h L_f \delta_f \frac{L_f^{n-1} - 1}{L_f - 1} + L_h L_f \delta_e (L_f^{n-1} - 1) \qquad = \delta + K_1 \qquad (24)$$

$$|\boldsymbol{s}_d[n] - \hat{\boldsymbol{s}}_d^{\text{ar}}[n]| \leqslant \delta + \delta \frac{L^n - L}{L - 1} \qquad\qquad\qquad = \delta + K_2 \qquad (25)$$

Now we consider two cases depending on the Lipschitz constants of the problem, namely $L_h, L_f$, and $L_e$. First, consider the case where the Lipschitz constants are very large (i.e. $L_h, L_f, L_e \gg 1$). In that case, the upper bounds can be approached by

$$K_1 \approx L_h \delta_f L_f^{n-1} + L_h L_f^n \delta_e \qquad (26)$$

$$K_2 \approx \delta_h (L_h L_f L_e)^{n-1} + L_h \delta_f L_f^{n-1} L_h^{n-1} L_e^{n-1} + L_h L_f^n \delta_e L_h^{n-1} L_e^{n-1} \qquad (27)$$

Hence, $K_2 \gg K_1$ (we highlighted the difference between both terms in the previous equation. Now consider the case where the Lipschitz constants are very small (i.e. $L_h, L_f, L_e \ll 1$). Recall that this case corresponds to a trivial prediction task since any trajectory of System 1 will converge to a unique state. Again, the upper bounds can be approached by

$$K_1 \approx 0 \qquad (28)$$

$$K_2 \approx L\delta \qquad (29)$$

In this trivial case, the upper bound on the prediction error using our method is a combination of the approximation errors from each function. On the other hand, using the classic AR scheme implies a larger error, since the model accumulates approximations at each time step not only from the dynamics but also from the observation function and the encoder.

## D   PROOF OF PROPOSITION 2

The proof follows the lines of Janny et al. (2022b). The existence of $\psi_q$ is granted by the observability assumption. Indeed assumption A2. states that for all $q > p$, $\mathcal{O}_q$ is injective in $\mathcal{S}$. Hence, it exists a inverse mapping $\mathcal{O}_q^* : \mathcal{O}_q : \mathcal{S} \mapsto \mathcal{S}$ such that $\forall \boldsymbol{s}' \in \mathcal{S}$

$$\mathcal{O}_q^* \big( \mathcal{O}_q(\boldsymbol{s}') \big) = \boldsymbol{s}' \qquad (30)$$

Let $\boldsymbol{z}_d[0|q] = \begin{bmatrix} \boldsymbol{z}_d[0] & \cdots & \boldsymbol{z}_d[q] \end{bmatrix}$. Hence, one can build $\psi_q$ using the dynamics of the system for all $\boldsymbol{x} \in \Omega$:

$$\forall \boldsymbol{s}_0 \in \mathcal{S}, \quad S(\boldsymbol{s}_0, \boldsymbol{x}, t) = S\Big( \mathcal{O}_q^* \big( \boldsymbol{z}_d[0|q] \big), \boldsymbol{x}, t \Big) := \psi_q \big( \boldsymbol{z}_d[0|q], \boldsymbol{x}, t \big) \qquad (31)$$

Now, because of the noise, the disturbed observation $\hat{\boldsymbol{z}}_d[0|q] = \boldsymbol{z}_d[0|q] + \delta_{0|q}$ may not belong to $\mathcal{O}_q(\mathcal{S})$, where the inverse mapping $\mathcal{O}_q^*$ is well defined. We solve this by finding the closest "possible" observation.

$$\hat{\boldsymbol{s}}_0 = \arg \min_{\boldsymbol{s}' \in \mathcal{S}} \big| \hat{\boldsymbol{z}}_d[0|q] - \mathcal{O}_q(\boldsymbol{s}') \big| \qquad (32)$$

$$\hat{\boldsymbol{s}}(\boldsymbol{x}, t) = S(\hat{\boldsymbol{s}}_0, \boldsymbol{x}, t) := \psi_q \big( \hat{\boldsymbol{z}}_d[0|q], \boldsymbol{x}, t \big). \qquad (33)$$

Hence, we have for all $\boldsymbol{s}' \in \mathcal{S}$

$$\Big| \hat{\boldsymbol{z}}_d[0|q] - \mathcal{O}_q \big( \hat{\boldsymbol{s}}_0 \big) \Big| \leqslant \Big| \hat{\boldsymbol{z}}_d[0|q] - \mathcal{O}_q \big( \boldsymbol{s}' \big) \Big|. \qquad (34)$$

In particular, for $\boldsymbol{s}' = \boldsymbol{s}_0$ and since $\mathcal{O}_q(\boldsymbol{s}_0) = \boldsymbol{z}_d[0|q]$,

$$\Big| \hat{\boldsymbol{z}}_d[0|q] - \mathcal{O}_q \big( \hat{\boldsymbol{s}}_0 \big) \Big| \leqslant \Big| \hat{\boldsymbol{z}}_d[0|q] - \mathcal{O}_q \big( \boldsymbol{s}_0 \big) \Big| \qquad (35)$$

$$\leqslant \big| \delta_{0|q} \big|.$$

In the other hand, from assumption A2. equation 8:

$$\alpha(p) |\hat{\boldsymbol{s}}_0 - \boldsymbol{s}_0|_{\mathcal{S}} \leqslant |\mathcal{O}_q(\hat{\boldsymbol{s}}_0) - \mathcal{O}_q(\boldsymbol{s}_0)| \qquad (36)$$

$$\leqslant |\mathcal{O}_q(\hat{\boldsymbol{s}}_0) - \hat{\boldsymbol{z}}_d[0|q]| + |\hat{\boldsymbol{z}}_d[0|q] - \mathcal{O}_q(\boldsymbol{s}_0)|$$

$$\leqslant 2 \big| \delta_{0|q} \big|$$

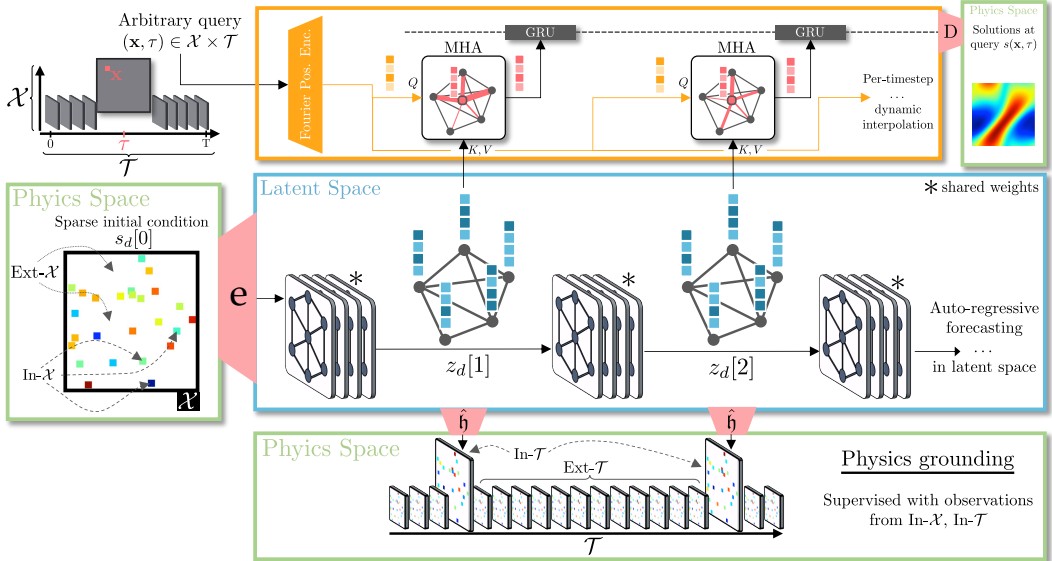

Figure 3: **Model overview** – The model leverages a dynamical system (System 1) to perform auto-regressive predictions of the dynamics in a mesh-structured latent space from sparse initial conditions. It is combined with a data-driven state estimator derived from another continuous-time dynamical system (System 2), implemented with multi-head cross-attention. The attention mechanism queries the intermediate anchor states from the auto-regressive predictor and uses Fourier positional encoding to encode the query points $(\mathbf{x}, \tau)$. An additional GRU refines the dynamics after interpolation.

Moreover, since $f_2$ is Lipschitz

$$\frac{\partial}{\partial t}|S(\boldsymbol{s}_0, \boldsymbol{x}, t) - S(\hat{\boldsymbol{s}}_0, \boldsymbol{x}, t)|_{\mathcal{S}} = |f_2\big(S(\boldsymbol{s}_0, \boldsymbol{x}, t)\big) - f_2\big(S(\hat{\boldsymbol{s}}_0, \boldsymbol{x}, t)\big)|_{\mathcal{S}} \tag{37}$$
$$\leqslant L_s |S(\boldsymbol{s}_0, \boldsymbol{x}, t) - S(\hat{\boldsymbol{s}}_0, \boldsymbol{x}, t)|_S.$$

and using the Grönwall inequality

$$|S(\boldsymbol{s}_0, \boldsymbol{x}, t) - S(\hat{\boldsymbol{s}}_0, \boldsymbol{x}, t)|_{\mathcal{S}} \leqslant e^{L_s t}|\boldsymbol{s}_0 - \hat{\boldsymbol{s}}_0|_{\mathcal{S}}. \tag{38}$$

Finally, combining equation 36 and equation 38

$$|S(\boldsymbol{s}_0, \boldsymbol{x}, t) - S(\hat{\boldsymbol{s}}_0, \boldsymbol{x}, t)|_{\mathcal{S}} \leqslant 2\alpha(q)^{-1}|\delta_{0|q}|e^{L_s t}.$$

which concludes the proof.

## E MODEL DESCRIPTION

In this section, we describe the architecture of our implementation in more detail. The model is illustrated in figure 3.

**Step 1** – The output predictor derived from System 1 is implemented as a multi-layer graph neural network inspired from Pfaff et al. (2020); Sanchez-Gonzalez et al. (2020) but without following the standard "*encode-process-decode*" setup. Let $\tilde{\mathcal{X}} = \{\boldsymbol{x}_0, ..., \boldsymbol{x}_K\}$ be the set of sub-sampled positions extracted from the known locations $\mathcal{X}$ (cf. Artificial generalization from section 3.4). The input of the module is the initial condition at the sampled points and the corresponding positions $\big(\boldsymbol{x}_i, \tilde{\boldsymbol{s}}_d[0](\boldsymbol{x}_i)\big)_i$ and is encoded into a graph-structured latent space $\boldsymbol{z}_d[0] = (\boldsymbol{z}_d[0]_i, \boldsymbol{e}[0]_{ij})_{i,j}$ where $\boldsymbol{z}_d[0]_i$ is a latent node embedding for position $\boldsymbol{x}_i$ and $\boldsymbol{e}[0]_{ij}$ is an edge embedding for edge pairs $(i, j)$ extracted from a Delaunay triangulation. The encoder $\hat{e}$ maps the sparse IC to node and edge embeddings using two MLPs, $f_{\text{edge}}$ and $f_{\text{node}}$:

$$\boldsymbol{z}_d[0]_i = f_{\text{node}}\big(\tilde{\boldsymbol{s}}_d[0](\boldsymbol{x}_i), \boldsymbol{x}_i\big), \quad \boldsymbol{e}_[0]ij = f_{\text{edge}}\big(\boldsymbol{x}_i - \boldsymbol{x}_j, |\boldsymbol{x}_i - \boldsymbol{x}_j|\big), \tag{39}$$

$f_{\text{nodes}}$ and $f_{\text{edges}}$ are two ReLU-activated MLPs, each consisting of 2 layers with 128 neurons. The initial node and edge features $z_d[0]_i$ and $e[0]_{ij}$ are represented as 128-dimensional vectors.

The dynamics $\hat{f}_1$ is modeled as a multi-layered graph neural network inspired from Pfaff et al. (2020); Sanchez-Gonzalez et al. (2020), we therefore add a layer superscript $^\ell$ to the notation:

$$z_d[n+1] = \hat{f}_1\big(z_d[n]\big) = \big(z_i^L, e_{ij}^L\big)_{i,j} \text{ such that } \begin{cases} e_{ij}^{\ell+1} &= e_{ij}^\ell + \overbrace{g_{\text{edge}}^\ell\big(z_i^\ell, z_j^\ell, e_{ij}^\ell\big)}^{\varepsilon_{ij}}, \\ z_i^{\ell+1} &= z_i^\ell + g_{\text{node}}^\ell\big(z_i^\ell, \sum_j \varepsilon_{ij}\big), \\ e_{ij}^0 &= e[n]_{ij}, \\ z_i^0 &= z_d[n]_i, \end{cases} \quad (40)$$

The GNNs employ two MLPs $g_{\text{node}}^\ell$ and $g_{\text{edges}}^\ell$ with same dimensions as $f_{\text{edges}}$ and $f_{\text{nodes}}$. We compute the sequence of anchor states $z_d[0], \cdots z_d[q]$ in the latent space by applying $\hat{f}_1$ auto-regressively.

The observation function $\hat{h}_1$ extracts the sparse observations $\tilde{s}_d[n]$ from the latent state $z_d[n]$ and consists of a two-layered MLP with 128 neurons, with Swish activation functions (Ramachandran et al., 2017) applied on the node features, i.e. $\tilde{s}_d[n](x_i) \approx \hat{h}_1\big(z[n]_i\big)$.

**Step 2** – The spatial and temporal domains $\Omega \times [\![0, T]\!]$ are normalized, since it tends to improve generalization on unseen locations. The state estimator $\psi_q$ takes as input the sequence of latent graph representation $z_d[0], \cdots, z_d[q]$ and a spatiotemporal query sampled in $\Omega \times [\![0, T]\!]$. This query is embedded in a Fourier space using the function $\zeta_\omega$ which depends on a frequency parameter $\omega \in \mathbb{R}^{\dim \Omega + 1}$ (initialized uniformly in $[0, 1]$). By concatenating harmonics of this frequency up to some rank, we obtain a resulting embedding of 128 dimensions (if $\zeta_\omega(x, t)$ exceeds the number of dimensions, cropping is performed to match the target shape).

$$\zeta_\omega(x, t) = [..., \cos(k\omega_{1|n_x}x), \sin(k\omega_{1|n_x}x), \cos(k\omega_{n_x+1}t), \sin(k\omega_{n_x+1}t), ...], \ k \in \{0, \cdots K\}. \quad (41)$$

The continuous variables $z_{n\Delta}(x, t)$ conditioned by the anchor states are computed with a multi-head attention Vaswani et al. (2017)

$$z_{n\Delta}(x, t) = f_{\text{mha}}\big(\text{Q}=\zeta_\omega(x, t), \text{K=V}=\{z_d[n]_i\} + \zeta_\omega(\mathcal{X}, n\Delta)\big), \quad (42)$$

where $f_{\text{mha}}$ is defined as

$$\begin{cases} q_1 &= A(Q, K, V), \\ q_2 &= Q + q_1, \\ q_3 &= B(q_2), \\ \text{out} &= q_3 + q_2. \end{cases} \quad (43)$$

Here, $A(\cdot, \cdot, \cdot)$ refers to the multi-head attention mechanism described in (Vaswani et al., 2017) with four attention heads, and $B(\cdot)$ represents a single-layer multi-layer perceptron activated by the rectified linear unit (ReLU) function. We do not use layer normalization.

The Gated Recurrent Unit Cho et al. (2014) aggregates the sequence of conditioned variables (of length $q$) as follows:

$$u[n] = r_{\text{gru}}\big(u[n-1], z_{n\Delta}(x, t)\big), \quad (44)$$

$$\hat{S}(s_0, x, t) = D\big(u[q]\big), \quad (45)$$

where $u[n]$ is the hidden memory of a GRU, initialized at zero. $r_{\text{GRU}}$ denotes the update equations of a GRU – we omit gating functions from the notation – and $D$ is a decoder MLP that maps the final GRU hidden state to the desired output, that is, the value of the solution at the desired spatiotemporal coordinate $(x, t)$, We used a two-layered gated recurrent unit with a hidden vector of size 128, and a two-layered MLP with 128 neurons activated by the Swish function for $D$.

**Training loop** – To create artificial generalization scenarios during training, we employ spatial sub-sampling. Specifically, during each gradient iteration, we randomly and uniformly mask 25% of $\mathcal{X}$ and feed the remaining 75% to the output predictor (System 1). To reduce training time further and improve generalization on unseen locations, we use bootstrapping by randomly sampling a smaller set of points for querying the model (i.e. as inputs to $\psi_q$). To do so, we maintain a probability weight vector $W$ of dimension $|\mathcal{X} \times \mathcal{T}|$, initialized to one. At each gradient descent step, we randomly select

$N=1,024$ points from $\mathcal{X} \times \mathcal{T}$ weighted by $W$. We update the weight matrix by setting the values at the sampled locations to zero and then adding the loss function value to the entire vector. This procedure serves two purposes: (a) it keeps track of poorly performing points (with higher loss) and (b) it increases the sampling probability for points that have been infrequently selected in previous steps.

The choice of $\Delta$ in the dynamics loss equation 13 allows us to reduce the complexity of the model. In Table 1, we present results obtained with $\Delta = 3\Delta^*$ indicating that the output predictor (System 1) predicts the latent state representation three time steps later. Consequently, the number of auto-regressive steps during training decreases from $T/\Delta^*$ (e.g., for MeshGraphNet and MAgNet) to $T/\Delta$. In Table 2, we used $\Delta = 2\Delta^*$. For a more comprehensive discussion on the effect of $\Delta$ on performances, please refer to Appendix G.

**Training parameters** – To be consistent, we trained our model with the same training setup over all different experiments (i.e. same loss function, and same hyper-parameters). However, for the baseline experiments, we did adapt hyper-parameters and used the ones provided by the original authors when possible (see further below). We used the AdamW optimizer with an initial learning rate of $10^{-3}$. Models were trained for 4,500 epochs, with a scheduled learning rate decay multiplied by $0.5$ after 2,500; 3,000; 3,500; and 4,000 epochs. Applying gradient clipping to a value of 1 effectively prevented catastrophic spiking during training. The batch size was set to 16.

# F   BASELINES AND DATASETS DETAILS

## F.1   BASELINES

The baselines are trained with the AdamW optimizer with a learning rate set at $10^{-3}$ for 10,000 epochs on each dataset. We keep the best-performing parameters on the validation set for evaluation on the test set.

**DINo** – we used the official implementation and kept the hyper-parameters suggested by the authors for *Navier* and *Shallow Water*. For Eagle, we used the same hyper-parameters as for *Shallow Water*. The training procedure was left unchanged.

**MeshGraphNet** – we used our own implementation of the model in PyTorch, with 8 layers of GNNs for *Navier* and *Shallow Water*, and up to 15 for *Eagle*. Other hyper-parameters were kept unchanged. We warmed up the model with single-step auto-regressive training with noise injection (Gaussian noise with a standard deviation of $10^{-4}$), as suggested in the original paper, and then fine-tuned the parameters by training on the complete available horizon. Both steps try to minimize the mean squared error between the prediction and the ground truth. Edges are computed using Delaunay triangulation. During evaluation, we perform cubic interpolation between time steps (linear interpolation gives better results on Eagle) first, then 2D cubic interpolation on space to retrieve the complete mesh.

**MAgNet** – We used our own implementation of the MAgNet[GNN] variant of the model, and followed the same training procedure as for MeshGrapNet. The parent mesh and the query points are extracted from the input data using the same spatial sub-sampling technique as in ours, and the edges are also computed with Delaunay triangulation. During evaluation, we split the query points into chunks of 10 nodes, and compute their representation with all the available measurement points. This reduces the number of interpolated vertices in the input mesh and improves performances at the cost of higher computation time (see figure 4). However, to be fair, this increase in computational complexity introduced by ourselves was not taken into account when we discussed computational complexity in appendix G.

## F.2   DATASET DETAILS

**Navier & Shallow Water** – Both datasets are derived from the ones used in (Yin et al., 2022). We adopted the same experimental setup but generated distinct training, validation, and testing sets. For details on the GT simulation pipeline, please see Yin et al. (2022). The *Navier* dataset comprises 256 training simulations of $40$ frames each, with additional two times 64 simulations allocated for validation and testing. Simulations are conducted on a uniform grid of 64 by 64 pixels (i.e. $\Omega$),

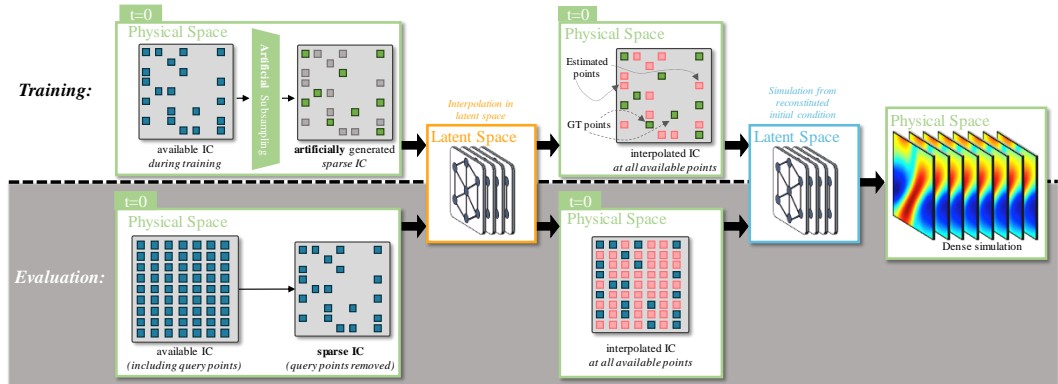

Figure 4: **MaGNet** – suffers from drastic shifts in distribution between training and evaluation. The model is trained on points from $\mathcal{X}$, which corresponds to a small portion of the domain. We used our subsampling trick to artificially generate queries. During evaluation, we require the prediction at every available point in the complete simulation, hence, MaGNet must interpolate the initial condition to a large number of query points, filling the input of the auto-regressive model with noisy estimates of the IC.

measuring the vorticity of a fluid subject to periodic forcing. During training, simulations were cropped to $T = 20$ frames. The *Shallow Water* dataset consists of 64 training simulations, along with 16 simulations in both validation and testing. Sequences of length $T = 20$ were generated. The non-euclidean sampling grid for this dataset is of dimensions $128 \times 64$.

**Eagle** – Eagle is a large-scale fluid dynamics dataset simulating the airflow generated by a drone within a 2D room. We extract sequences of length $T = 10$ from examples within the dataset, limiting the number of points to 3,000 (vertices were duplicated when the number of nodes fell below this threshold).

The spatially down-sampled versions of these datasets (employed in Table 1 and 2) were obtained through masking. We generate a random binary mask, shared across the training, validation, and test sets, to remove a specified number of points based on the desired scenario. Consequently, the observed locations remain consistent across training, validation, and test sets, except Eagle, where the mesh varies between simulations. For *Navier* and *Shallow Water*, the *High* setup retains $25\%$ of the original grid, the *Middle* setup retains $10\%$, and the *Low* setup retains $5\%$. In the case of Eagle, the *High* setup preserves $50\%$ of the original mesh, while the *Low* setup retains only $25\%$. Temporal down-sampling was also applied by regularly removing a fixed number of frames from the sequences, corresponding to no down-sampling ($1/1$ setup), half down-sampling ($1/2$), and quarter down-sampling ($1/4$). During evaluation, the models are tasked with predicting the solution to every location and time instant present in the original simulation.

## G MORE RESULTS

**Time continuity** – is illustrated in Figure 6 on the *Navier* dataset. Our model and the baselines are trained in a very challenging setup, where only part of the information is available. During training, not only does the spatial mesh only contains $25\%$ of the complete simulation grid, but also the time-step is increased to four time its initial value. In this situation, the model needs to represent low-resolution data while being trained on sparse data.

**Generalization to unseen future timesteps** – Beyond time continuity, our model offers some generalization to future timesteps. Table 3 shows extrapolation results for high/mid/low subsampling of the spatial data on the *Navier* dataset which outperforms the predictions of competing baselines.

**Generalization to unseen grid** – In our spatial and temporal interpolation experiments (tables 1 and 2 of the main paper), we assumed that the observed mesh remains identical during training and testing. Nevertheless, the ability to adapt to diverse meshes is an important aspect of the task. To evaluate this capability, we trained our model in the spatial extrapolation setup on the Navier dataset.

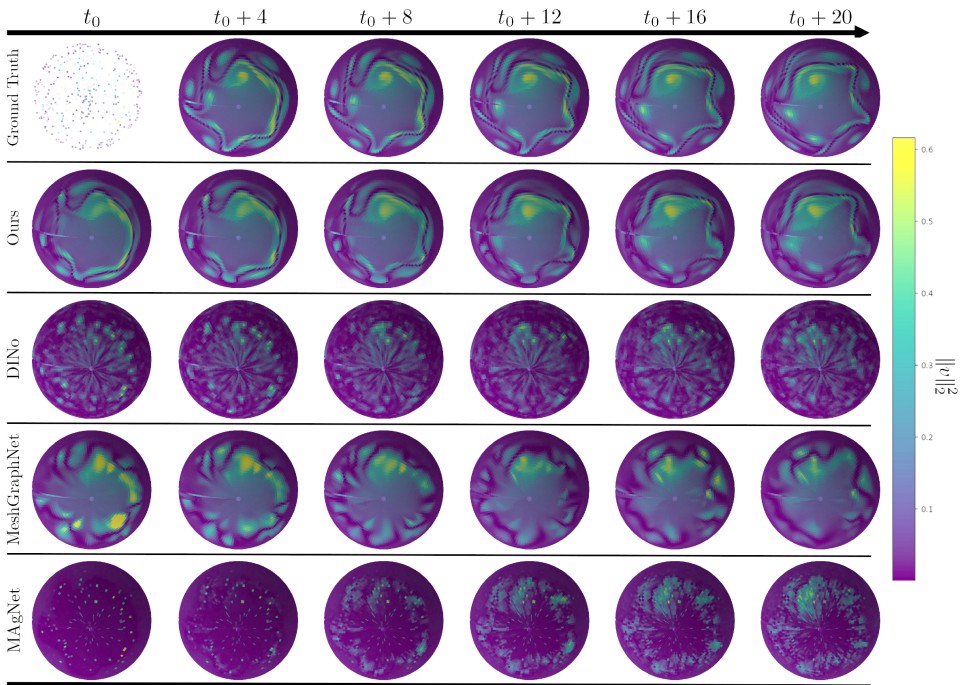

Figure 5: **Qualitative results on Shallow-Water** – Simulation obtained with our model and the baseline in the challenging 5% setup on the *Shallow Water* dataset (without temporal sub-sampling). Each model is initialized with a small set of sparse observations and needs to extrapolate the solution at many unseen positions. Our model outperforms the baselines, which struggle to compute the solution outside the training domain.

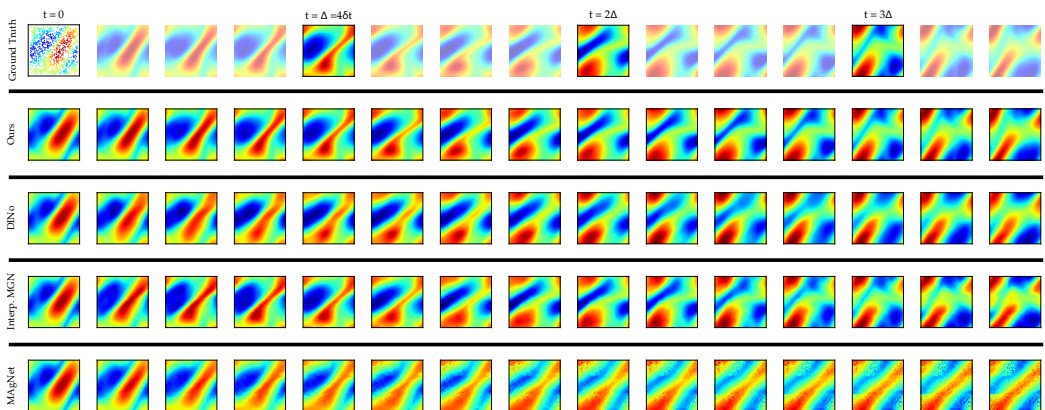

Figure 6: **Time continuity on the *Navier* dataset** – during training, models are only exposed to a sparse observation of the trajectories, represented spatially by the dots in the upper left figure and temporally by the semi-transparent frames. Our model maintains the temporal consistency of the solution and outperforms the baselines.

We compute the error when exposed to different meshes, potentially with a different sampling rate, and report the results in table 4. Our model demonstrates good generalization skills when confronted with new and unseen grids. We observe that the error on new grids is close to the error reported in table 1 in the Ext-$\mathcal{X}$ case, we show additionally that the model can generalize even if the observed grid is different. Notably, the model performs well when trained with a medium sampling rate. Despite some performance degradation when the evaluation setup is significantly different compared

|  |  | *Navier* | | |
|---|---|---|---|---|
|  |  | High | Mid | Low |
| DINo | In-$\mathcal{X}$ | 2.266 | 2.017 | 3.154 |
|  | Ext-$\mathcal{X}$ | 2.317 | 2.136 | 6.740 |
| Interp. MGN | In-$\mathcal{X}$ | 6.853 | 3.136 | 1.378 |
|  | Ext-$\mathcal{X}$ | 7.632 | 6.890 | 15.55 |
| MAgNet | In-$\mathcal{X}$ | 171.5 | 31.07 | 10.02 |
|  | Ext-$\mathcal{X}$ | 227.0 | 57.60 | 89.20 |
| **Ours** | In-$\mathcal{X}$ | **0.3732** | **0.3563** | **0.3366** |
|  | Ext-$\mathcal{X}$ | **0.3766** | **0.3892** | **0.6520** |

Table 3: **Time Extrapolation** – We assessed the performances of our model vs. the baselines in a time-extrapolation scenario by forecasting the solution on a horizon two times longer than the training one (i.e. 40 frames). Our model remains more performant.

|  |  |  | Training | | | | | |
|---|---|---|---|---|---|---|---|---|
|  |  |  | Navier | | | Shallow | | |
|  |  |  | High | Mid | Low | High | Mid | Low |
| Evaluation | High | In-$\mathcal{X}$ | **0.2492** | 0.7929 | 4.5165 | **0.5224** | 1.5431 | 4.3447 |
|  |  | Ext-$\mathcal{X}$ | **0.2477** | 0.7782 | 4.4038 | **0.5256** | 1.5822 | 4.4963 |
|  | Mid | In-$\mathcal{X}$ | 0.4370 | **0.3230** | 0.9759 | 0.8528 | **1.2908** | 3.6766 |
|  |  | Ext-$\mathcal{X}$ | 0.4410 | **0.3401** | 0.9496 | 0.8617 | **1.2589** | 3.6043 |
|  | Low | In-$\mathcal{X}$ | 2.2000 | 0.4039 | **0.6732** | 2.4395 | 1.5634 | **3.4793** |
|  |  | Ext-$\mathcal{X}$ | 2.2037 | 0.4216 | **0.7892** | 2.3914 | 1.5313 | **3.2334** |

Table 4: **Generalization to unseen grid** – We investigate generalization to previously unseen grids by training our model on the Navier dataset in the space extrapolation setup. We report the error (MSE ($\times 10^{-3}$)) inside and outside the spatial domain $\mathcal{X}$ measured with different sampling rates unseen during training. The diagonal shows results on grids with identical sampling rates wrt. training, but sampled differently. Our model shows great generalization properties.

to training, our model effectively maintains its interpolation quality between out-of-domain error (Ext-$\mathcal{X}$) and in-domain error, testifying to the robustness of our dynamic interpolation module.

**Ablations** – we study the impact of key design choices in Figure 7a. First, we show the effect of the subsampling strategy to favor learning of spatial generalization, c.f. Section 3.4, where we subsample the input to the auto-regressive backbone by keeping 75% of the mesh. We ablate this feature by training the model on 100%, 50%, and 25% of the input points. When the model is trained on 100% of the mesh, it fails to generalize to unseen locations, as the model is always queried on points lying in the input mesh. However, reducing the number of input points significantly further from the operating point decreases the performance of the backbone, as it does not dispose of enough points to learn meaningful information for prediction. We also replace the final GRU with simpler aggregation techniques, such as a mean and a maximum pooling, which drastically degrades the results. Finally, we ablate the dynamics part of the training loss (Eq. 13). As expected, this deteriorates the results significantly.

**More ablation on the interpolator** – We conducted an ablation study to show that limiting attention is detrimental. To do so, we designed four variants of our interpolation module:

- **Single attention (w/o GRU)** – performs the attention between the query and the embeddings in a single shot, rather than time-step per time-step. This variant neglects the insights from control theory presented in section 3.1 (Step 2). The single softmax function limits the attention to a handful of points, whereas our method encourages the model to attend to at least one point per time step and reason on a larger timescale, considering past and future predictions, which is beneficial for interpolation tasks, as supported by proposition 2.

- **Spatial (w/ GRU) & Temporal (w/o GRU) neighborhood** – limit the attention to the nearest temporal or spatial points, which significantly degrades the metrics. To handle setups with sparse and subsampled trajectories, the interpolation module greatly benefits from not only distant points but also from the temporal flow of the simulation.

|  | Ours | Single attention | Temporal attention | Spatial neigh. | Temporal neigh. | ANP Kim et al. (2018) |
|---|---|---|---|---|---|---|
| In-$\mathcal{X}$ / In-$\mathcal{T}$ | **0.2113** | 0.3863 | 0.2912 | 0.5623 | 0.4130 | 1.734 |
| Ext-$\mathcal{X}$ / In-$\mathcal{T}$ | **0.2251** | 0.4168 | 0.3180 | 0.6328 | 0.6681 | 1.835 |
| In-$\mathcal{X}$ / Ext-$\mathcal{T}$ | **0.2235** | 0.4094 | 0.3095 | 0.6030 | 1.9624 | 1.820 |
| Ext-$\mathcal{X}$ / Ext-$\mathcal{T}$ | **0.2371** | 0.4388 | 0.3350 | 0.6741 | 2.1818 | 1.920 |

Table 5: **Ablation on interpolation** – We performed four ablations on the interpolation module (MSE ($\times 10^{-3}$)). *Single attention* combines all $z_d[n]$ into a single key vector, employing attention only once (w/o GRU). *Temporal attention* replaces the GRU with a 2-head attention, *Spatial neigh.* restricts attention to the five spatially nearest points from the query, and *Temporal neigh.* computes attention only with the nearest time $z_d[n]$ to the queried time $\tau$ (w/o GRU). These results indicate that considering long-range spatial and temporal interactions is beneficial for the interpolation task.

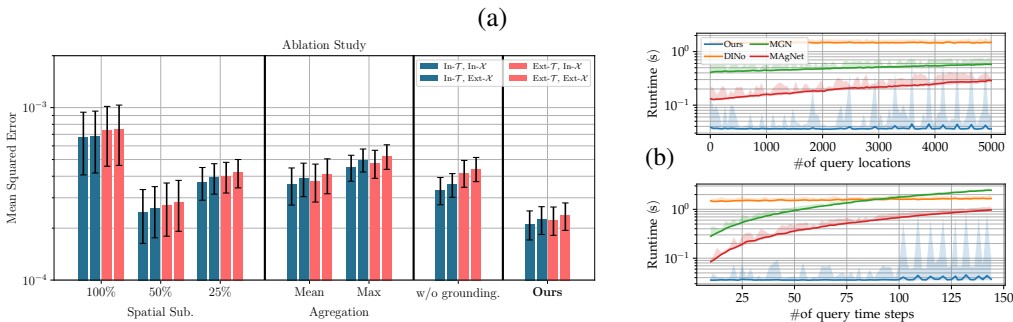

Figure 7: **Ablations and runtime** – (a) Ablations on Navier (Yin et al., 2022; Stokes, 2009) with 10% of data and half temporal resolution, from left to right: exploring subsampling strategies, replacing GRU par mean/max pooling, removing physics grounding. (b) Runtime analysis as a function of query locations and time steps, respectively. The graph shows the average runtime (over 100 runs), and shadows indicate lower and upper bounds over the runs.

- **Temporal attention (w/o GRU)** replaces the GRU in our model with a 2-head attention layer. This variant of our model does not improve the performance compared to a GRU. We argue that GRU is more suited for accumulating observations in time, as its structure matches classic observer designs in control theory.

- **Attentive Neural Process** Kim et al. (2018) is a interpolation module close to ours resembling the *Single attention* ablation, with an additional global latent $c$ to account for uncertainties. The model involves a prior function $q(c, s)$ trained to minimize the Kullback-Leibler divergence between $q\big(z, s(\mathcal{X}, \mathcal{T})\big)$ (computed using the physical state at observed points) and $q\big(c, s(\Omega \setminus \mathcal{X}, [\![0, T]\!])\big)$ (computed using the physical state at query points).

Results are shown in table 5. All ablations exhibit worse performance than ours. Note that the ANP ablation involves performing the interpolation in the physical space to compute the Kullback-Leibler divergence during training. Thus, the interpolation module cannot use the latent space from the auto-regressive module, which may explain the drop in performance. Adapting ANP to directly leverage the latent states is probably possible, but not straightforward and requires several key changes in the architecture.

**Efficiency** – the design choices we made led to a computationally efficient model, compared to prior work. For all three baselines, the required number of computed time steps for the auto-regressive rollout depends on (1) the number of predicted time steps, and (2) the time values themselves, as for later values of $t$, more iterations need to be computed. In contrast, our method forecasts using attention from a set of "anchor states", which is controlled through the hyper-parameter $\Delta$. The length of the auto-regressive rollout is therefore constant and does not depend on the number of predicted time steps. Furthermore, while DINo scales very well to predict additional locations, it requires a costly optimization step to compute $\alpha_0$. MGN does benefit from the efficient cubic interpolation algorithm, which is a side effect of the fact that it has been adapted to this task, but not

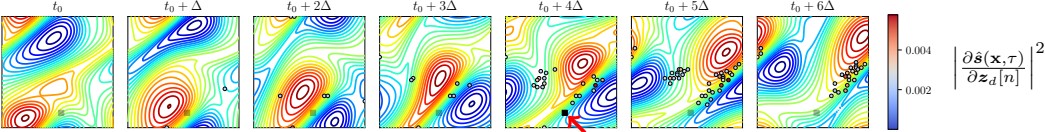

(a) **Frontier tracking**: when queried on a streamline between areas of opposite vorticity, the interpolation module attends not only to the spatial neighborhood but also to the temporal flow near the frontier.

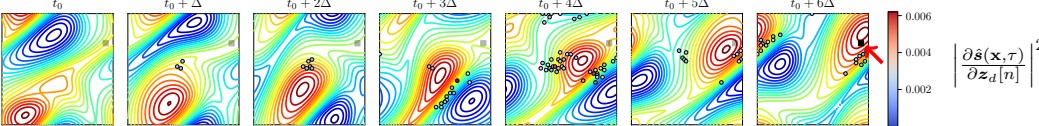

(b) **Blob tracking**: in homogeneous areas, the model tracks the origin of the perturbation, and focuses on its displacement. Our dynamic interpolation exploits the evolution of the state rather than merely averaging neighboring nodes.

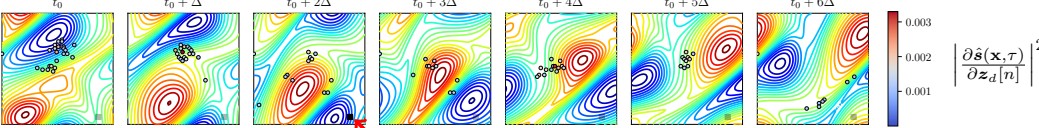

(c) **Periodic boundaries**: our model effectively leverages the periodic condition of the Navier dataset, especially when queried on points originating from perturbations on the other side of the simulation. Again, the interpolation depends on which points explain the output, rather than the neighborhood.

Figure 8: **Norm of output derivative** – wrt. each $z_d[n](x_i)$ (Navier, high spatial subsampling setup). We display top-100 nodes (●) with the highest norm, i.e. the most important nodes for interpolation at query point (■). Using gradients rather than attention allows us to visualize the action of the GRU. We observe context-adaptive behaviors, leveraging temporal flow information over local neighbors, challenging to implement in handcrafted algorithms.

designed for it. We experimentally confirm these claims in Figure 7, where we provide the evolution of runtime as a function of query locations, and of query time steps, respectively. In both cases, our model compares very favorably to competing methods.

**Attention maps** – To further support our claims, we analyzed the behavior of the interpolation module in more depth and showed the top-100 most important nodes from the embedding points $z_d[n](x_i)$ used to interpolate at different queries. The figure is shown in Figure 8. We observed very complex behaviors that dynamically adapt to the global situation around the queried points. Our interpolation module appears to give more importance to the flow rather than merely averaging the neighboring nodes, thus relying on "why" the queried point is in a specific state. Such behavior would be extremely difficult to implement in a handcrafted algorithm.

**Parameter Sensitivity Analysis** – We investigate the influence of two principal hyper-parameters, namely the step size $\Delta$ and the number of residual GNN layers $L$, on the performances of our model. We present the results of our experiments in figure 9 on the *Navier* dataset, which has been spatially down-sampled at 10% during training and has a temporal resolution reduced by two.

The choice of the step size between iterations of the auto-regressive backbone directly affects both training and inference time. For a trajectory of $T$ frames, the number of anchor states $z_d[n]$ is determined by $\lfloor T/\Delta \rfloor$. Increasing the step size $\Delta$ of the learned dynamics leads to a higher number of embeddings over which the models need to reason. A parallel can be drawn between this phenomenon and the influence of the discretization size on the accuracy of numerical methods for solving PDEs. Furthermore, the selection of $\Delta$ also impacts the generalization capabilities of the model in Ext-$\mathcal{T}$. When $\Delta > \Delta^*$, the model is queried during training on intermediate instants not directly associated with any of the anchor states $z_d[n]$. This is visible in Figure 9 where, for instance, with $\Delta = \Delta^*$, the In-$\mathcal{X}$/In-$\mathcal{T}$ error is the lowest, but other metrics increases compared to $\Delta = 2\Delta^*$.

The number of layers $L$ in the auto-regressive backbone significantly influences the overall performance of the model, both within the domain and on the exteriors. Increasing the number of layers generally leads to improved performance. However, it appears that beyond $L = 8$, the error starts to

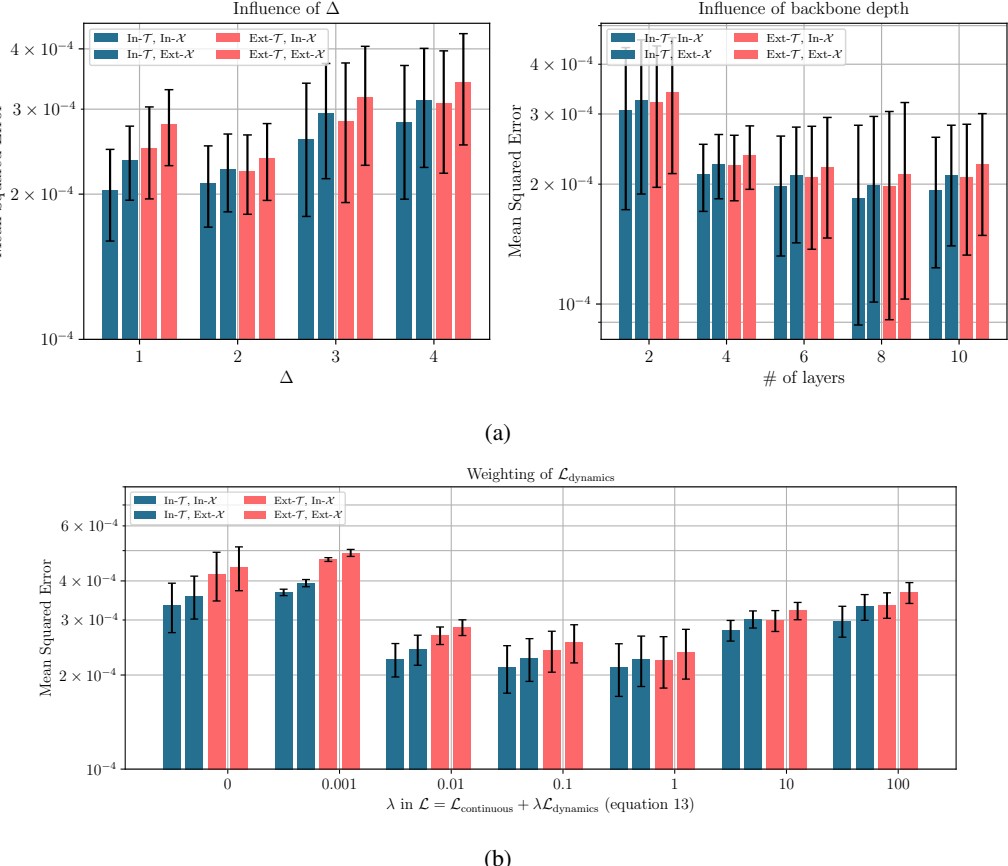

(a)

(b)

Figure 9: **Impact of hyper-parameters on model performance** – We evaluate the impact of three critical hyper-parameters on our architecture, namely, (a) the step size $\Delta$, the depth $L$ of the auto-regressive backbone and (b) the weighting of the dynamics cost in equation 13. To assess the performance, we employ the 10% *Navier* dataset with $1/2$ frames and compute metrics for both in-domain and out-domain. The results reveal that increasing the depth of the GNN layers enhances the model's performance, while lower values of $\Delta$ lead to better metrics. However, we observed a degradation in the ability of the model to generalize to unseen time instants for the special case $\Delta=\Delta^*$. Moreover, we found that equally weighting of both terms $\mathcal{L}_{\text{continuous}}$ and $\mathcal{L}_{\text{dynamics}}$ leads to best results.

increase, indicating a saturation point in terms of performance gain. The relationship between the number of layers and model performance is visually depicted in Figure 9. Throughout this paper, we maintain this hyper-parameter constant for the sake of simplicity, as our primary focus is the spatial and temporal generalization of the solution.

**Failure cases** – we expose failure cases on the Eagle dataset (in the *Low* spatial down-sampling scenario) in figure 10. In some particularly challenging instances of this turbulent dataset, we noticed drops in accuracy located in fast-evolving regions of the simulation, and in particular near the flow source. We hypothesize that the origin of the failure is related to the comparatively smaller processor unit used in our auto-regressive backbone compared to the baseline introduced in Janny et al. (2023), hence producing less accurate anchor states when the horizon increases.

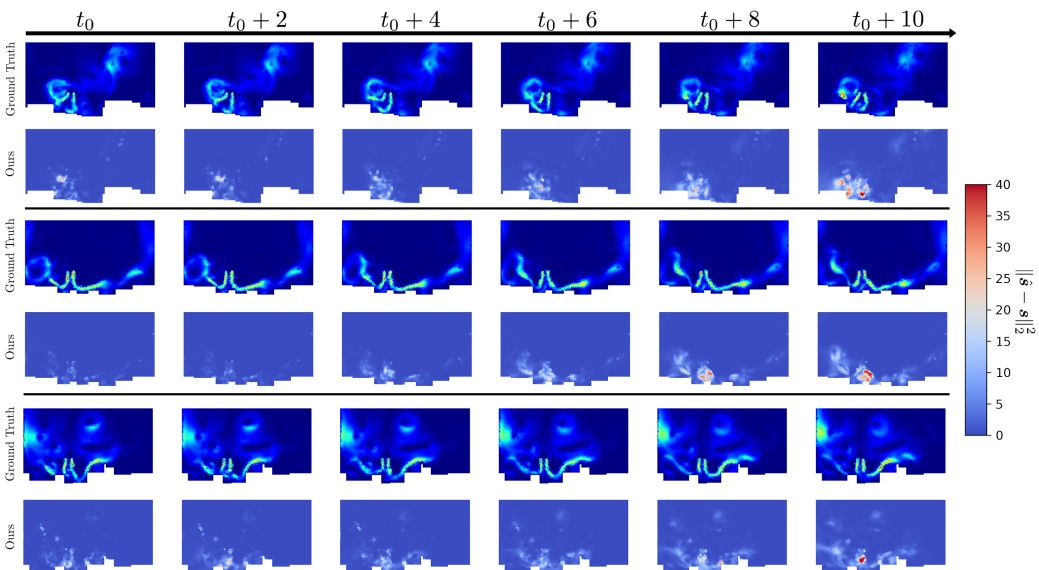

Figure 10: **Failure cases on Eagle**– We observed failure cases on highly challenging instances of the EAGLE dataset as the prediction horizon is increasing. We show the per point error in three different instances and observed that the error increases with the time horizon, especially close to turbulent areas, such as below the UAV.

