# OpenReview forum: "Space and time continuous physics simulation from partial observations"
_ICLR.cc/2024/Conference — ICLR 2024 spotlight_

### Official Review · Reviewer_VJPj · 2023-10-30

**Soundness:** 4 excellent
**Presentation:** 3 good
**Contribution:** 3 good
**Rating:** 8
**Confidence:** 3

**Summary:**

The paper addresses the problem of limitations in traditional physical simulations, which rely on numerical schemes and mesh-refinement methods to balance precision and complexity. The proposed methodology is based on a double observation challenge, which involves extracting regularities from weakly informative physical variables that are sparsely measured in space and time. The authors propose a solution with two interlinked dynamical systems defined on, respectively, the sparse positions and the continuous domain. System 1 is a discrete dynamical model used to compute a sequence of latent anchor states z_d auto-regressively, while System 2 is used to design a state estimator q retrieving the dense physical state at arbitrary locations (x, t). The proposed methodology has several potential implications and effectiveness. First, it allows for predictions in a continuous spatial and temporal domain, which is not possible with traditional methods. Second, it is trained on sparse observations, making it highly adaptable and efficient. Third, the proposed method is generalizable and can be applied to a wide range of physical systems.

**Strengths:**

1. The proposed methodology represents a significant departure from traditional physical simulations and has the potential to improve the efficiency and accuracy of physical simulations.
2. The authors demonstrate the effectiveness of the proposed methodology through experiments on three datasets in fluid dynamics and compare their approach to strong baselines, which are outperformed both in classical settings and in the extended new task requiring continuous predictions.
3. The proposed method is generalizable and can be applied to a wide range of physical systems.
4. The authors provide a website with an online interactive tool to visualize the results.
5. The paper is well-organized and clearly presents the proposed methodology and experimental results.

**Weaknesses:**

1. The paper outlines the utilization of a dynamical system for auto-regressive predictions in a mesh-structured latent space, starting from sparse initial conditions. However, it lacks a comprehensive description of the neural network architecture employed. Clarification is needed on how the model systematically handles and compensates for missing data, as this is a crucial aspect for the practical application and reliability of the proposed method.

2. While the document mentions the use of a specific optimization algorithm for training the model, it omits essential details like the learning rate schedule and batch size. A more thorough examination and justification of the chosen hyperparameters, including regularization parameters and the number of hidden layers, is necessary to ensure replicability and to provide insights into the model's design choices.

3. The paper criticizes traditional physical simulations based on partial differential equations (PDEs) and numerical algorithms for their limitations in complex phenomena and handling sparse observations. It is imperative to acknowledge the extensive success and development of these traditional methods across various applications over the years. Data-driven methods, despite showing potential in overcoming some of these limitations, bring their own set of challenges and limitations. These include the substantial data requirements for training and the often opaque nature of these models, which can make interpretation and validation of results challenging. It would be beneficial for the authors to provide a balanced view, comparing the strengths and weaknesses of both approaches to give readers a comprehensive understanding of the context in which their work sits.

**Questions:**

1. The authors do not provide a thorough comparison of their approach to other data-driven methods in the literature. Can you provide a more detailed comparison of your approach to other data-driven methods in the literature, and how your approach differs from these methods?

2. The paper could benefit from a more detailed analysis of the computational requirements of the proposed methodology. Can you provide more information on the computational requirements of your approach, and how it compares to traditional physical simulations in terms of computational efficiency?

3. The authors do not provide a detailed analysis of the interpretability of the proposed model. Can you provide more information on the interpretability of your approach, and how you plan to address the challenge of interpretability in data-driven methods?

4. The paper could benefit from a more detailed discussion of the potential implications and applications of the proposed methodology beyond the specific examples presented in the paper. Can you provide more information on the potential implications and applications of your approach in other physical systems beyond fluid dynamics?

---

> ### Author Response · Authors · 2023-11-20
> **Reply**
>
> Thank you for your review.
>
> > **The paper outlines the utilization of a dynamical system for auto-regressive predictions in a mesh-structured latent space, starting from sparse initial conditions. However, it lacks a comprehensive description of the neural network architecture employed. Clarification is needed on how the model systematically handles and compensates for missing data, as this is a crucial aspect for the practical application and reliability of the proposed method.**
>
> An in-depth description of the architecture is available in Appendix E of the paper. The insights supporting our design choices are explained in section 3. Please let us know if any information is missing, we will gladly update the manuscript.
>
> > **While the document mentions the use of a specific optimization algorithm for training the model, it omits essential details like the learning rate schedule and batch size. A more thorough examination and justification of the chosen hyperparameters, including regularization parameters and the number of hidden layers, is necessary to ensure replicability and to provide insights into the model's design choices.**
>
> All the details for reproducibility are available in Appendix E, including optimization details. You will find an extensive set of additional results in Appendix G that better support our model, including ablations and influence studies of some key hyper-parameters.
>
> > **The paper criticizes traditional physical simulations based on partial differential equations (PDEs) and numerical algorithms for their limitations in complex phenomena and handling sparse observations.**
>
> Of course, we do not claim that our method is a drop-in replacement solution to conventional solvers, and apologize if our paper suggests the opposite. We updated the introduction of the paper to clarify that conventional methods are still the main method for simulation.
>
> > **The authors do not provide a thorough comparison of their approach to other data-driven methods in the literature.**
>
> We compare our method with state-of-the-art approaches and oracles in the experiment section of the paper. We also included theoretical comparisons in section 3.2 (paragraph Discussion and related work).
>
> > **The authors do not provide a detailed analysis of the interpretability of the proposed model.**
>
> Interpretability is indeed difficult to assess with data-driven methods. Yet, we showed attention maps in Fig. 8 of the appendix. Our experiment highlights intricate behaviors for the interpolation module which benefits not only from the local neighborhood of the query point but also from more distant interactions.
>
> > **The paper could benefit from a more detailed discussion of the potential implications and applications of the proposed methodology beyond the specific examples presented in the paper.**
>
> Thank you, we extended our conclusion to mention potential applications to different physical phenomena.

---

### Official Review · Reviewer_fAqF · 2023-10-30

**Soundness:** 3 good
**Presentation:** 3 good
**Contribution:** 3 good
**Rating:** 8
**Confidence:** 3

**Summary:**

The paper proposes a data-driven and spatiotemporal continuous surrogate model that generalizes to new unseen initial conditions of partial differential equations. The key idea of the paper is the use of double observable spaces, in which continuous response can be computed on top of learned latent dynamical systems. Training objective is defined on both of the observable spaces. The proposed method is compared with strong baselines in representative experiments.

**Strengths:**

The paper is appropriately placed in the current growing literature on scientific machine learning to create a novel spatiotemporal continuous model for tackling complex and wide range of problems in the realm of PDEs. The exposition for the motivation is written crisply and is generally easy to follow. The proposed method is also analyzed from the theoretical aspect and the experiments strengthen the validity of the proposed method.

**Weaknesses:**

What the authors mean by “Generalize to new unseen initial conditions (ICs)” in R2 is a it unclear. Does this mean initial conditions are even ouf of distribution?


It’s still a bit difficult to assess the degree of the paper’s contribution since the authors miss some literatures relevant to the method proposed in Step1, which are so called “latent unroll” [1, 2]. Although those papers do not provide theoretical results, the idea is essentially same (except the format of input) and its effectiveness is shown empirically. The authors might want to refer those papers in the right place and clarify the contribution of the paper.


[1] "Learning to Accelerate Partial Differential Equations via Latent Global Evolution," T. Wu, T. Maruyama, J. Leskovec. NeurIPS, 2022.\
[2] “Learning latent field dynamics of PDEs,” A. Sanchez, D. Kochkov, J. A. Smith, M. Brenner, P. Battaglia, and T. J. Pfaff,  NeurIPS, 2020.

The spacial resolution of the encoder's input in the most experiences (and the number of subsequent samples) is low. It is concerning if the proposed method is scalable to large resolution since $f_1$ is implemented as MPNN that hardly captures global information unless the number of nodes is small. Do you observe any relationship among the number of processor blocks of $f_1$, the spacial resolution, and the resulting error?




**Minor comments**:

* Why does the variance of the runtime of proposed method in Figure 7(b) sporadically get high?

* Typo: "r1, r2 et r3" in the last paragraph of the section 1.

**Questions:**

See the weakness above.

**Details Of Ethics Concerns:**

No concerns.

---

> ### Author Response · Authors · 2023-11-20
> **Reply**
>
> Thank you for your review.
>
> > **What the authors mean by “Generalize to new unseen initial conditions (ICs)”**
>
> The statement is indeed ambiguous. We clarify the main paper by rephrasing R2 as “The method must be capable of handling new initial conditions that do not belong explicitly to the training set, without re-training or fine-tuning.”
>
> > **The authors might want to refer those papers in the right place and clarify the contribution of the paper.**
>
> Thank you for pointing out these missing references, they have been added to the manuscript. However, we highlight that these papers address a very different task, and our contribution is not limited to the latent unroll method. In particular:
> - We support our design with theoretical results allowing us to analyze why our approach is beneficial,
> - We address the task of continuous production by introducing a new interpolation module based on attention,
> - We outperform the state-of-the-art, including on challenging datasets such as Eagle,
> - We provide numerous ablation studies and experiments to better support our method.
>
> > **The spacial resolution of the encoder's input in the most experiences (and the number of subsequent samples) is low. It is concerning if the proposed method is scalable to large resolution since f1 is implemented as MPNN that hardly captures global information unless the number of nodes is small. Do you observe any relationship among the number of processor blocks of f1, the spacial resolution, and the resulting error?**
>
> This is an interesting and debatable remark. We would like to make the following observations:
> 1) The low spatial resolution is motivated by our task, which seeks generalization over arbitrary positions, including unobserved ones. In the case of dense observation, there is probably no need for our method.
> 2) GNNs are indeed known to struggle to capture long-range information. However, several empirical results (in particular, MeshGraphNet, Pfaff et al, 2021) show that they succeed in forecasting the dynamics of complex physics phenomena with dense observation grids.
> 3) We conducted a study of the influence on the number of processor blocks of $f_1$ in Fig. 9a of the appendix and show that more blocks tend to improve the performances up to some extent.
>
> Finally, we argue that most of our contributions (ie. theoretical results, interpolation module based on attention, etc.) still hold with a different architecture of the processor unit, including more advanced GNN structures that may fit a particular problem.
>
> > **Why does the variance of the runtime of proposed method in Figure 7(b) sporadically get high?**
>
> Figure 7(b) shows the evolution of the runtime for a varying number of query positions (top graph) and query timesteps (bottom graph). The graphs show the runtime averaged on 100 consecutive runs of each model, and we represent the lower and upper bounds in the envelope (not the variance, this has been made explicit in the paper). We hypothesize that these spikes are due to outliers in the runs, potentially either due to the PyTorch or the python runtime engine. However, we observed that, in average, our method is sensibly faster than the baselines.

---

> ### Comment · Reviewer_fAqF · 2023-11-21
> **Thank the authors for the reply.**
>
> Thank you for your answers. Most of my concerns were addressed and the novelty of the paper is much clearer. The theoretical and empirical results support the claim that the proposed method can meet the 3 requirements. Given the efforts on clarification of the details and the contributions of the paper, I would recommend the paper for acceptance and am raising my score to 8.

---

### Official Review · Reviewer_Kpnn · 2023-11-01

**Soundness:** 4 excellent
**Presentation:** 4 excellent
**Contribution:** 3 good
**Rating:** 8
**Confidence:** 3

**Summary:**

This paper presents a data-driven framework that leverages an auto-regressive structure combined with transformer inference, GNN, and GRU techniques to predict continuous spatial and temporal dynamics in fluid systems from sparse observations. The proposed model demonstrates the ability of generalization to unseen initial conditions.

**Strengths:**

- The paper is well-motivated and well structured
- The proposed approach is novel to the best of my knowledge, particularly in dividing the dynamics into different systems
- Including theoretical analysis pertaining to error bounds on two systems enhances the paper's soundness - this approach not only solidifies the work but also offers valuable insights
- Experiment visualizations are compelling (with nice work on the website), highlighting the advantages of continuous prediction results against relevant baselines in scientific computing

**Weaknesses:**

- The model's reliance on a regular sampling rate could constrain its applicability. In many practical scenarios, we might not have access to a dataset $\mathcal{D}$ with a consistent sampling rate. This could impact model training, given that the anchor states are determined based on the available rate.
- A discussion on how the anchor state computation rate $\Delta$ influences performance would be beneficial. Given that fluid dynamic systems can exhibit high-frequency changes, a delayed computation rate might fail to capture these rapid transitions.


---

This does not influence my score: what is the meaning of the small Figure on page 9 on the right? Is it a mistake?

**Questions:**

- For System 2, the use of GRU implies iterative discrete predictions rather than time-continuous ones. Does this allow for predictions at arbitrary time frames?
- If the sampling rate $\Delta$ does not align with $\Delta^*$ such that $\Delta$ is not an integer multiple of $\Delta^*$, would this lead to enhanced model generalization for unseen time intervals?
- I am curious about the performance over long-time rollouts. Does executing rollouts in the hidden space contribute to more stable long-term predictions?
- When the query position $(x, t)$ has a time $t$ that falls between two anchor states, how does the model do inference?

---

> ### Author Response · Authors · 2023-11-20
> **Reply**
>
> Thank you for your review.
>
> > **The model's reliance on a regular sampling rate could constrain its applicability.**
>
> We indeed acknowledge this limitation in the conclusion of the main paper. Handling non-uniform sampling times is a more challenging task, and requires redesigning the auto-regressive forecaster. This modification is not trivial and requires additional work. In particular, our theoretical framework needs to be extended to a continuous setup for System 1. However, we believe that this is one of the next milestones for the future of our approach, and will explore the possibility of re-designing System 1 as a continuous-time system (implemented with Neural-ODE), thus allowing us to handle non-uniform time steps.
>
> > **A discussion on how the anchor state computation rate Δ influences performance would be beneficial.**
>
> This study is available in Fig. 9b of the supplementary material. We observed that a higher value of $\Delta$ degrades performances (since System 2 benefits from less anchor to interpolate the solution at an arbitrary location). However, choosing $\Delta=\Delta^*$ leads to poor generalization capacity in the time domain.
>
> > **what is the meaning of the small Figure on page 9 on the right? Is it a mistake?**
>
> The small figure is a teaser of the experiments conducted in the supplementary material since many ablations are presented in the appendices.
>
> > **For System 2, the use of GRU implies iterative discrete predictions rather than time-continuous ones. Does this allow for predictions at arbitrary time frames?**
>
> There is a misunderstanding. As stated in eq. (44) (45), the GRU is used to aggregate estimations of the solution (at arbitrary time and space locations) performed at different anchor points. As their name suggests, the “*anchor points*” provide estimates at fixed anchors which provide computational support, but they do not correspond to the time instant for which the prediction will be made. We designed the GRU to aggregate information over these anchors, but prediction and interpolation are performed by the GNNs and the attention module. Our model can perform predictions at arbitrary time frames, as shown in Table 2 (see *Ext-T* metrics)
>
> > **If the sampling rate Δ does not align with Δ∗ such that Δ is not an integer multiple of Δ∗, would this lead to enhanced model generalization for unseen time intervals?**
>
> The value of the sampling rate $\Delta$ is limited by the need for supervision of the auto-regressive backbone (we showed in Figure 7 of the appendix that removing this supervision is detrimental). If $\Delta$ is not an integer multiple of $\Delta^*$, we do not have access to supervision for the auto-regressive backbone.
>
> > **I am curious about the performance over long-time rollouts. Does executing rollouts in the hidden space contribute to more stable long-term predictions?**
>
> As stated in the main paper, time extrapolation is outside the scope of this paper, but we agreed that this is an interesting property, and performed an extrapolation experiment reported in Table 3 of the appendix. We show that our model is capable of longer horizon prediction, and still outperforms the baselines.
>
> > **When the query position (x,t) has a time t that falls between two anchor states, how does the model do inference?**
>
> The key to arbitrary time inference is the use of the attention mechanism to compute an estimation of the state $s(x,t)$ (with arbitrary position and time) conditioned by the anchor state, ie. $z_{n\Delta}$ in eq. 42 of the main paper. These estimations are then merged using the GRU. It is important to note that, while the anchor points are discrete, the model can still interpolate to arbitrary positions thanks to the attention module.

---

> > ### Comment · Reviewer_Kpnn · 2023-11-22
> >
> > Thank you for your comments. I have no additional concerns. I will keep my score of 8; I believe this paper will positively impact this area.

---

### Official Review · Reviewer_eMKs · 2023-11-06

**Soundness:** 3 good
**Presentation:** 4 excellent
**Contribution:** 3 good
**Rating:** 6
**Confidence:** 2

**Summary:**

This paper proposes a way to perform forward modeling given sparse initial measurements and is able to retrieve dense solutions out of it. To do so, this paper proposes two systems. One system takes sparse measurement as input - encodes into latent space and propagates in the latent space. Another system takes propagating latent space features and predicts the dense solution. It shows that it works better under sparse observation situations.

**Strengths:**

Solving an interesting problem where there are only sparse observations, which is common in real-world scenarios. The overall formulation seems novel and there are some interesting theoretical derivations.

**Weaknesses:**

It is not clear how the intrinsic ill-posedness of the system is addressed - since there is only sparse observation given in the initial state, if i understand correctly.

**Questions:**

1. in the webpage - 2D Navier dataset visualization - what does it mean by "supervision". Does it mean that the supervision used during the training is those sparse observations in space over time? If the training data is that sparse, how does a data-driven approach learn high-resolution details?

2. as discussed in the paper, sparse observation yields an ill-posed state except the observed time period is long enough. How would the author resolve the ill-posedness? Are there any failure modes observed during the test time?

3. would it be possible to incorporate initial observation over a longer time step (instead of just s_d[0], but maybe s_d[0],s_d[1],...,s_d[n]) to make the problem less ill-posed. Given such a sparse initial state, there could be many plausible initial states, I am curious why would this work...

---

> ### Author Response · Authors · 2023-11-20
> **Reply**
>
> Thank you for your review.
>
> > **in the webpage - 2D Navier dataset visualization - what does it mean by "supervision". Does it mean that the supervision used during the training is those sparse observations in space over time? If the training data is that sparse, how does a data-driven approach learn high-resolution details?**
>
> The term “*supervision*” in the webpage indeed indicates the supervision used during training, ie. the sparse observation in the training dataset. Yet, while humans are not capable of reconstructing the high-resolution simulation, data-driven methods can exploit the physical prior knowledge extracted from the (large-scale) dataset during training to reconstruct missing parts using “physics-guided” inference. To give a very high-level explanation,
> - the model learns regularities in two different ways: (1) For each instance (trajectory) it receives sparse observations over time, from which it can infer temporal regularities; (2) is also observes a large amount of trajectories from which it can infer regularities on full dataset/problem level by fusing all sparse observations and linking them to their initial conditions. This is a hard problem, but there is a large amount of information in the data.
>
> - as other inverse problems, the  interpolation of sparse observations is a problem which is traditionally less well solved by humans than computational models, be they based on precise modeling and inverting, or data-driven. The human visual system has not been “trained” for this kind of problems, whereas computational models can pick up and exploit regularities efficiently, if they exist.
>
> > **as discussed in the paper, sparse observation yields an ill-posed state except the observed time period is long enough. How would the author resolve the ill-posedness? Are there any failure modes observed during the test time?**
>
> As for many problems in machine learning, the Bayes error for this problem is not zero, i.e. there are instances for which it cannot be solved fully, but this does not mean that good quality predictions cannot be made by picking up regularities from data. As a similar case we can cite the SfM (structure from motion) problem in computer vision, which is known to be theoretically solvable only up to scale, but data-driven methods can produce unnormalized predictions by picking up regularities from data (eg. “a car is not 3km long”).
>
> In our case, since the ground-truth PDE is unknown, addressing the ill-posedness nature of the problem fully theoretically is difficult. However, it can be mitigated using observability considerations. For instance, our method uses a sequence of discrete states to infer the state at arbitrary locations rather than a single time frame. Moreover, sampling points are distributed uniformly to prevent unobservable areas, which could lead to ill-posedness. In a nutshell, as many (most?) inverse problems solved with data-driven methods, we cannot address the issues fully theoretically, but argue that our model is tailored through inductive biases to mitigate it.
>
> We would even go further and argue, that data-driven have a particular potential in the cases, where the illposed-ness nature of a problem is hard to model, and it is in these cases where they beat model-based methods.
> We exposed and discussed failure modes on the EAGLE dataset in Appendix G of the paper.
>
> > **would it be possible to incorporate initial observation over a longer time step (instead of just s_d[0], but maybe s_d[0],s_d[1],...,s_d[n]) to make the problem less ill-posed. Given such a sparse initial state, there could be many plausible initial states, I am curious why would this work...**
>
> This could be indeed possible and is supported by the concept of finite-time observability of dynamical systems (which is a milder assumption compared to strict observability) but requires access to a sequence of initial conditions. This is not the case in most simulation applications, which are initialized with a unique IC.
>
> Furthermore, using a sequence would require observing or simulating more time steps, before using our model to perform prediction hence lowering the need for a prediction.

---

### Official Review · Reviewer_uqXM · 2023-11-09

**Soundness:** 3 good
**Presentation:** 4 excellent
**Contribution:** 3 good
**Rating:** 8
**Confidence:** 4

**Summary:**

In this paper, the authors presented a neural-network-based model that predicts the evolution of a physical system, starting from sparse observations. In order to do so, they consider two (Markovian) dynamical systems (one discrete and one continuous) to predict the time evolution and query the solution in unseen points in the domain.

**Strengths:**

The methodology is well presented and the claims are supported by analytical proofs. The proposed model does provide better results then the models available in the literature, both in time and space

**Weaknesses:**

While the method works well in practice for the presented datasets, I think that the authors could highlight better the hypothesis behind their model. On the one hand, they correctly highlight how DINo (Yin et al. 2022) assumes the existence of a learnable ODE that models the system dynamics. On the other hand, the authors' approach hypothesizes the existence of a Markovian discrete system that is able to represent the advance in time. This is an equivalently strong hypothesis in my opinion. Furthermore, we assume that an latent space $\mathbf{z}_d$ can be found using an encoder. This is even more relevant since they perform the time advancement in the latent space: with a scheme "encode-process-decode" at each timestep, the encoder needs to take care only of the spatial representation, instead in this case the temporal dynamics also need to be embedded.

**Questions:**

- I would consider adding a bit more information about the weaknesses highlighted before, while they do not change the results, they can provide a more clear picture of the limitations of the model
- In the introduction, the model from Yin et al. (2022) does satisfy R3, so it would be better to correct the statement there
- The loss function (eq. 13) has two terms, did you check how the balance of the two terms affects the solution?
- The author state that there is a trade-off in the performance of $\phi_q$. Did you train different models to quantify it? What is the dependence of the performance on the training sequence division (q) and length (T). How is it related with the timescales of the physical phenomena tested?

- minor: typo in fig.1, where $\phi_q$ is $\phi_p$

As a final observation, the model performs well but in my opinion the tasks considered are a bit simplistic and do not really highlight the limitations of the proposed models.
It would be better to focus on more challenging benchmarks (for instance Biferale et al. https://arxiv.org/abs/2006.07469) in order to bring these models closer to actual applications.

---

> ### Author Response · Authors · 2023-11-20
> **Reply**
>
> Thank you for your review.
>
> > **I think that the authors could highlight better the hypothesis behind their model.**
>
> We added clarification in the discussion in section 3.2. Apart from learnability, our approach leverages 3 hypotheses:
> 1) **The existence of a discrete System 1** can be interpreted as a requirement on the sparse observations: we require the observed samples to follow a dynamics, ie. to contain enough information to be predictable, although it can be potentially complex. Markovianity is a mild assumption when applied to physical systems, but may not exist for some edge cases.
> 2) **The observability of System 1** (ie. the existence of an encoder) this hypothesis (shared with DINo) allows us to retrieve a complete representation of the state of the system from an observation. This is mandatory since we perform prediction from a single-frame sparse initial condition rather than a sequence.
> 3) **Finite-time observability of System 2** is discussed thoroughly in the paper and is arguably a milder assumption compared to the strict observability condition required in MaGNeT
>
> With respect to Dino, one difference which we think might by important is the fact that our dynamics is supposed to be in an embedding space and not in the space of network parameters (Dino predicts modulations for implicit representations).
>
> > **the encoder needs to take care only of the spatial representation, instead in this case the temporal dynamics also need to be embedded.**
>
> The temporal dynamics is indeed embedded in the latent space $z_d$, but this does not affect the encoder, which is only used once on the initial condition. Temporality will be handled by the (much more expressive) GNNs layers. Moreover, we argue that, since the state representation remains in the latent space, our model can store and use temporal knowledge in the vector. On the other end, encode-process-decode schemes discard this information to fall back to the physical space.
>
> > **In the introduction, the model from Yin et al. (2022) does satisfy R3, so it would be better to correct the statement there**
>
> We apologize for this confusion and fixed the introduction in the main paper. Fortunately, this error did not propagate, and DINo is correctly presented in the rest of the paper.
>
> > **The loss function (eq. 13) has two terms, did you check how the balance of the two terms affects the solution?**
>
> We conducted an ablation study by zeroing the term related to System 1 in the loss and showed that this impacts performances negatively. In practice, we observed that equal weighting of each term already results in significantly outperforming the baselines. We show the impact of the weighting in the cost function in new figure 9b in the appendix.
>
> > **The author state that there is a trade-off in the performance of psi_q. Did you train different models to quantify it?**
>
> We did investigate the relationship between the accuracy of the model and the length of the anchor sequence by testing different values of $\Delta$ (see Fig. 9a). We indeed observe that reducing the number of anchor states (ie. increasing $\Delta$) has a negative impact on performance, as suggested by Proposition 2. Yet, we also observed that exploiting every observed time frame (ie. $\Delta$=$\Delta^*$) is detrimental to the interpolation capacity of the model.
>
> > **It would be better to focus on more challenging benchmarks**
>
> While we agree that one of the next milestones of our approach is to address more and more challenging tasks, we would like to point out that the tasks used in our experiments are standard and widely used in very recent literature (in particular Yin et al. 2022). In addition to the benchmarks covered in this paper, we also conducted experiments on the challenging EAGLE dataset, which represents a leap in difficulty with regards to the literature.

---

### Author Response · Authors · 2023-11-20
**General Reply**

We would like to thank the reviewer for the time and work they spent on their reviews, we appreciate the feedback. We are glad they appreciate the quality of our work and in particular the soundness of our theoretical results. They also highlight the novelty of the approach, the quality of the experiment, and the clarity of the manuscript.

Following the suggestions, we updated our manuscript to improve clarity and to provide some more requested details. These updates are highlighted in blue in the rebuttal pdf. Moreover, as proposed by the reviewers, we added two new figures in the supplementary material:
- Fig 9b shows the impact of the weighting between each term in the training cost (eq.13)
- Fig 10 shows failure cases on the Eagle Dataset.

We hope that our updates on the paper address the questions of the reviewers.

---

### Meta-Review · Area_Chair_zHdL · 2023-12-10

**Metareview:**

The paper has received positive feedback from five knowledgeable reviewers with directly relevant experiences in the area. The AC believes the contribution, regarding architecture design and formal analyses, are clear and significant when it comes to the continuous interpolation, in the latent space, from sparse measurements while being trained only using sparse measurements and the final implementation achieves quite good performance compared to a couple of prior recent works on a few datasets of different difficulties. The contributions are unanimously acknowledged by the reviewers.


There were also some initial questions and concerns regarding the realisticness of the benchmarks, precision of the writing regarding prior work and method description, ablation studies, training setup, ill-position of the problem, uniformity of the observation rate, the capability of the architecture to model long-range dependencies, and a proper acknowledgement of the conventional less data-driven techniques.


The AC believes the revised paper as well as the public rebuttal addresses those concerns that could affect the outcome of the paper. Therefore, the AC suggests acceptance.

**Justification For Why Not Higher Score:**

The paper is almost equally qualified to be presented as an oral paper. A minor point is a few concerns regarding the realisticness of the various setups and benchmarks as outlined in the original meta review, without which it would have been a clear candidate for oral presentation.

**Justification For Why Not Lower Score:**

The paper has clear and significant contributions for an important problem that has real world impact and is rapidly growing in interest from the ML audience. It certainly deserves a spotlight presentation.

---

### Decision · Program_Chairs · 2024-01-16

Accept (spotlight)